# Securing the Language of Life: Inheritable Watermarks from DNA Language Models to Proteins

**Zaixi Zhang**[*]
Princeton University
zz8680@princeton.edu

**Ruofan Jin**
Zhejiang University
ruofanjin@zju.edu.cn

**Le Cong**[*]
Stanford University
congle@stanford.edu

**Mengdi Wang**[*]
Princeton University
mengdiw@princeton.edu

## Abstract

DNA language models have revolutionized our ability to understand and design DNA sequences—the fundamental language of life—with unprecedented precision, enabling transformative applications in therapeutics, synthetic biology, and gene-editing. However, this capability also poses substantial dual-use risks, including the potential for creating pathogens, viruses, even bioweapons. To address these biosecurity challenges, we introduce two innovative watermarking techniques to reliably track the designed DNA: DNAMark and CentralMark. DNAMark employs synonymous codon substitutions to embed watermarks in DNA sequences while preserving the original function. CentralMark further advances this by creating inheritable watermarks that transfer from DNA to translated proteins, leveraging protein embeddings to ensure detection across the central dogma. Both methods utilize semantic embeddings to generate watermark logits, enhancing robustness against natural mutations, synthesis errors, and adversarial attacks. Evaluated on our therapeutic DNA benchmark, DNAMark and CentralMark achieve F1 detection scores above 0.85 under various conditions, while maintaining over 60% sequence similarity to ground truth and degeneracy scores below 15%. A case study on the CRISPR-Cas9 system underscores CentralMark's utility in real-world settings. This work establishes a vital framework for securing DNA language models, balancing innovation with accountability to mitigate biosecurity risks.

## 1 Introduction

DNA serves as the cornerstone of the central dogma [13], orchestrating the flow of genetic information from DNA to RNA to proteins. Within this paradigm, DNA encodes the genetic blueprint, RNA acts as a dynamic messenger, and proteins execute a vast array of cellular functions (Figure 1 a). Recent advances in DNA language models have transformed our ability to understand and design DNA sequences with unprecedented precision [46, 8, 74, 47, 71, 42]. These models leverage computational frameworks to decode complex sequence patterns, enabling groundbreaking applications in therapeutics, synthetic biology, gene-editing, and beyond.

However, the remarkable capabilities of DNA language models also introduce significant dual-use risks [7, 52, 5, 76, 23]. For example, these models could lower the barrier to the creation of harmful biological agents, such as pathogens, viruses, or bioweapons. State-of-the-art DNA models excel in predicting and generating sequences with missense mutations or pathogenic properties

---

[*]Corresponding Authors

39th Conference on Neural Information Processing Systems (NeurIPS 2025).

[46, 8, 71, 42, 18], amplifying biosecurity concerns. The AI and scientific communities have recognized the emerging risks of DNA language models and are advocating robust guardrails and comprehensive oversight mechanisms [70, 63, 5, 52, 49].

Recently, watermarking has emerged as an effective strategy to counter the misuse of large language models (LLMs), enabling the traceability of generated content to ensure accountability and mitigate risks such as misinformation or malicious output [16, 34]. However, the application of watermarking to DNA language models presents unique and underexplored challenges. Unlike LLMs, which operate on expansive vocabularies, DNA language models are constrained by a *small alphabet* of only four nucleotides, complicating the design of robust watermarking strategies, such as green/red list approaches. Moreover, DNA is susceptible to *natural mutations [62], synthesis errors, and sequencing inaccuracies* [58], which can obscure or degrade watermarks. Additional complexities arise from biological constraints to preserve the *functional integrity* of encoded sequences to maintain their utility in applications like protein engineering. These challenges necessitate new watermarking frameworks tailored to the biological and computational intricacies of DNA sequence design.

To tackle these challenges, we propose a function-invariant watermark **DNAMark** using synonymous codon substitutions and **CentralMark** that builds an inheritable watermark transferable from designed DNA to translated protein. DNAMark and CentralMark address the challenges with the following innovations: **(1)** To achieve robust watermark resistant to natural mutations and potential attacks, DNAMark and CentralMark utilize the generated DNA or translated protein embeddings (Evo2 [8] or ESM [37]) to predict watermark logits with trained watermark models. The watermark logits are then added to the original logits from DNA models to bias the next nucleotide selection for watermarking. The intuition is that DNA and protein embeddings are inherently robust to minor mutations, preserving semantic and functional integrity during watermark logit prediction. During training, the watermark model is optimized to prioritize semantic preservation and maintain an unbiased distribution, enhancing watermark robustness and performance. **(2)** To minimize disruption to DNA sequence quality and encoded protein function, **DNAMark** employs a sparse watermarking scheme with synonymous codon substitutions, selectively modifying only the third base of specific codons to ensure the resulting codon encodes the same amino acid as the unmarked sequence (Figure 1 d). **(3)** To ensure inheritable watermark in both DNA and translated protein, **CentralMark** predicts watermark logits from protein embeddings and applies the watermark to the second base of each codon, enabling near non-overlapping separation of amino acids into green/red lists, facilitating reliable watermark detection across the central dogma (Figure 1 e).

Using our curated therapeutic DNA benchmark (Figure 1b), **DNAMark** and **CentralMark** achieve robust F1 detection scores (>0.85) under various attacks, including nucleotide substitution, insertion, and deletion attacks. Meanwhile, DNA sequence qualities are preserved, with over 60% sequence similarity to ground truth and degeneracy scores below 15%. Case studies on watermarking a CRISPR-Cas9 system [11, 12] designed by Evo model [46] (Figure 1c) demonstrate **CentralMark**'s potential for practical applications in real-world synthetic biology and gene-editing.

## 2 Related Works

### 2.1 Watermark for Language Models

Driven by the need to identify machine-generated text and mitigate potential misuse, the field of watermarking large language models (LLMs) has seen rapid development. Early and influential approaches, such as the one proposed by Kirchenbauer et al. [34], often referred to as KGW, introduced a method of biasing token generation towards a "green list" determined by a pseudorandom function seeded by preceding tokens. This creates a statistical watermark detectable with high accuracy (More details in Section 3). Subsequent works have aimed to improve detectability [24, 40, 35], text quality [30, 25, 28, 72], capacity [24, 69, 73], robustness [39, 53], and public verifiability [22, 38]. For Example, to enhance watermark detectability, EWD [40] assigns weights to tokens based on their entropy during detection, enhancing sensitivity by emphasizing high-entropy tokens in z-score calculations. To mitigate the logits bias brought by KGW applying a uniform $\delta$ to green list tokens, Hu et al. [30] introduced two unbiased reweighting methods to preserve the original text distribution. Aiming at increasing the watermark capacity to convey additional information like timestamps, identifiers, or copyright. Fernandez et al. [24] expand binary vocabulary partition to multi-color partition. To further improve watermark robustness against removal attacks such as

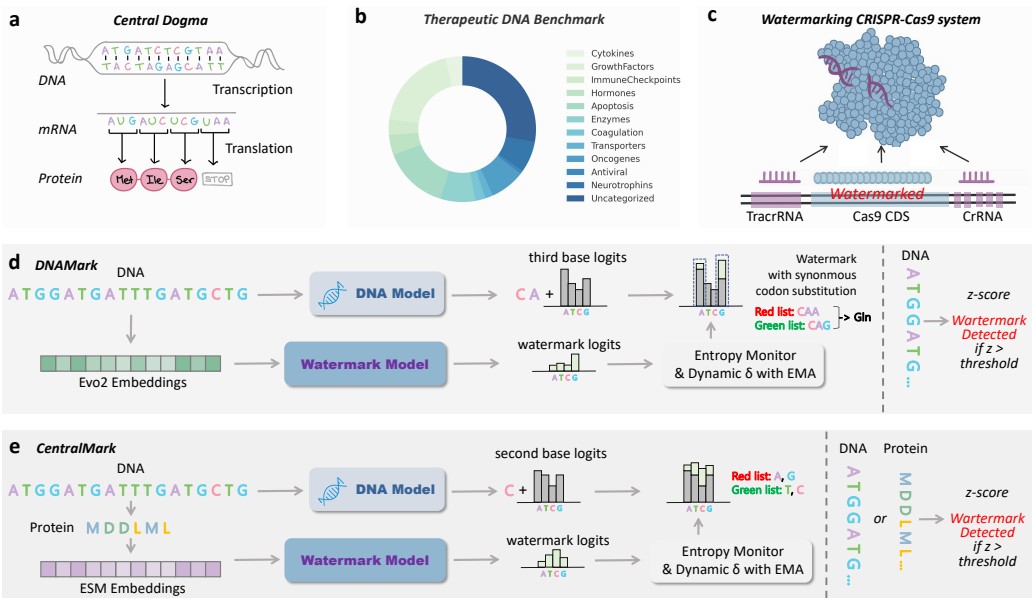

Figure 1: Watemark DNA language models with DNAMark and CentralMark. (a). DNA plays a key role in the central dogma; (b). A therapeutic DNA benchmark is constructed to evaluate DNA watermarks; (c) Our watermark methods successfully watermarks CRISPR-Cas9 generated by Evo; (d) DNAMark leverages watermark models and synonymous codon substitutions for DNA watermark; (e) CentralMark uses ESM-based watermark model to achieve an inheritable watermark. Watermark can be detected in both the DNA and the translated protein sequence generated with CentralMark.

paraphrasing, semantic-invariant watermark methods [39, 53] are proposed to ensure that similar text semantics result in similar partition outcomes, which are robust to attacks. To achieve publicly verifiable watermarks, Fairoze et al. [22] have utilized a digital signature technology from the field of cryptography, involving generating watermarks using a private key and verifying them with a public key. Recently, Zhang et al. [75] and Chen et al. [9] applied watermarks to protein generative models. However, it is unknown whether a watermark scheme can be designed for DNA language models and the central dogma.

## 2.2 DNA Language Models

Driven by advances in LLMs, DNA Language Models (DNA LMs) have also experienced rapid progress in recent years. Early DNA LMs primarily focused on DNA sequence interpretation and property prediction [33, 78, 55, 4]. For instance, Enformer combined convolutional down-sampling with transformer layers to enable accurate gene-expression prediction [4], while the Nucleotide Transformer, trained on multi-species corpora, markedly improved variant-effect prediction [15]. More recently, DNA LMs with advanced sequence generation capabilities have emerged [59, 74, 47, 71, 42, 46, 8]. For example, HyenaDNA leveraged implicit long-range convolutions to scale context to one million tokens [47]. GENERATOR, a 1.2B transformer decoder trained on 386 billion base pairs of eukaryotic DNA, excels in generating viable protein-coding sequences [71]. Evo, a 7B model trained on billions of prokaryotic and viral base pairs, demonstrated advanced capabilities in designing CRISPR–Cas complexes [46]. Its successor, Evo2, was scaled using 9.3 trillion DNA base pairs with one-million-token context windows, yielding autoregressive models with 7B and 40B parameters. Evo2 enables genome-wide prediction and *de novo* synthesis of DNA sequences across all domains of life [8]. Evo2 excels in generating chromosome-scale sequences, including similar sequences to human mitochondrial, *M. genitalium*, and *S. cerevisiae* genomes.

The advanced capabilities of DNA language models simultaneously raise significant biosafety and biosecurity concerns [70, 63]. Current countermeasures, such as sequence screening [1] and regulatory policies [5], are often suboptimal, as they may fail to detect AI-generated sequences or adapt to

evolving model capabilities [49]. Robust watermarking techniques tailored for DNA could enable reliable tracing and detection of AI-generated DNA sequences, addressing these gaps.

# 3   Preliminaries

Autoregressive language models, such as transformer-based architectures, generate text by modeling the conditional probability of a token given its preceding context. Formally, for a sequence of tokens $\mathbf{x} = (x_1, x_2, \ldots, x_T)$, an autoregressive model predicts the next token $x_t$ based on the probability distribution $p(x_t|x_{1:t-1}; \theta)$, where $\theta$ denotes the model parameters. The joint probability of the sequence is expressed as:

$$p(\mathbf{x}; \theta) = \prod_{t=1}^{T} p(x_t|x_{1:t-1}; \theta). \tag{1}$$

These models excel at producing coherent and contextually relevant text, but their widespread use raises concerns about content authenticity, ownership, and traceability.

To address these challenges, watermarking techniques embed imperceptible identifiers into the outputs of language models. A watermark is a subtle, structured modification to the generated text, designed to be robust against post-processing (e.g., paraphrasing) while remaining inconspicuous to human readers. For example, the KGW watermarking scheme [34] modifies the token probability distribution during generation. Specifically, for a vocabulary $\mathcal{V}$, KGW partitions tokens into a "green" list $\mathcal{G} \subset \mathcal{V}$ and a complementary "red" list $\mathcal{R} = \mathcal{V} \setminus \mathcal{G}$ based on a cryptographic hash of the context. The probability of selecting a token $x_t \in \mathcal{G}$ is boosted by an additive term $\delta$, altering the sampling distribution as:

$$p_{\text{wm}}(x_t|x_{1:t-1}; \theta) \propto p(x_t|x_{1:t-1}; \theta) + \delta \cdot \mathbb{I}(x_t \in \mathcal{G}), \tag{2}$$

where $\mathbb{I}(\cdot)$ is the indicator function, and the modified distribution is normalized. This ensures the watermark is embedded without significantly degrading text quality.

Watermark detection involves identifying the presence of these embedded identifiers in a suspect text. In the KGW scheme, detection leverages a statistical hypothesis test based on the z-score, which quantifies the likelihood that a given text $\mathbf{x}$ was generated by a watermarked model. Specifically, the detector counts the number of tokens in the green list, denoted $r = \sum_{t=1}^{T} \mathbb{I}(x_t \in \mathcal{G})$, over the sequence of length $T$. Under the null hypothesis (no watermark), tokens are sampled uniformly from $\mathcal{V}$, and the expected proportion of green tokens is $\gamma = |\mathcal{G}|/|\mathcal{V}|$. The z-score is computed as:

$$z = \frac{r - \mathbb{E}[r]}{\sqrt{\text{Var}[r]}} = \frac{r - T \cdot \gamma}{\sqrt{T\gamma(1 - \gamma)}}, \tag{3}$$

where $\mathbb{E}[r] = T \cdot \frac{|\mathcal{G}|}{|\mathcal{V}|}$ and $\text{Var}[r] = T\gamma(1 - \gamma)$ assume a binomial distribution for $r$. A high z-score (e.g., $z \geq \tau$ for a threshold $\tau$) indicates the presence of the watermark, as the observed green token count significantly exceeds the expected count under the null hypothesis.

# 4   Methods

## 4.1   DNAMark: Function-invariant Watermark for DNA Models

To achieve resistance to natural mutations and function preservation for synthetic biology, we first build DNAMark (Figure 1 (d)), a robust, and function-invariant watermark scheme for DNA language models in this section. Inspired by previous works on semantic-invariant watermarks for LLMs [39, 53], DNAMark utilizes **a specialized trained watermark model to generate watermark logits** for robustness. For watermarking in the coding region, we use **synonymous codon substitutions** to keep the coded amino acid unchanged. Moreover, **adaptive watermark strength** and **entropy-guided watermark strategy** are applied to balance sequence quality and detection accuracy.

### 4.1.1   Watermark Model based on Evo2 Embeddings

To embed a robust watermark in generated DNA sequences, DNAMark processes the sequence preceding the current token through the Evo2 [8] model to obtain functional embeddings, which are

then transformed into watermark logits and combined with the original token logits. Leveraging DNA's inherent robustness as an information carrier [10, 26, 21], where small mutations typically preserve encoded biological functions, DNAMark is designed to provide a durable watermark for DNA language models, resisting both natural mutations and adversarial modifications. Specifically, the watermark model in DNAMark satisfies two critical properties: *semantic preservation*, ensuring the watermark maintains the sequence's biological semantics (e.g., protein coding or regulatory roles) by aligning logit similarities with Evo2 embedding similarities. Moreover, the logits should be varied sufficiently to enhance complexity and security. Otherwise, if the watermark logits are monotonous, the green list is more static and might be revealed by counting the token frequency. This compromises the watermark protection and leads to the risk of being cracked. The second property, *unbiased distribution*, ensures that watermark logits exhibit no systematic preference for any nucleotide or codon and maintain a balanced distribution of positive and negative values, enhancing security against statistical attacks and ensuring robust, detectable watermarks for DNA sequences.

To realize these properties, we trained the watermark model [39] (Appendix. G), comprising multiple fully connected layers and layer norm, with two main loss functions: an alignment loss and a normalization loss. The alignment loss aligns the watermark logit similarity with the Evo2 embedding similarity: we normalize the embedding similarities by subtracting their mean and applying the hyperbolic tangent function. The alignment loss $\mathcal{L}_a$ is defined as:

$$\mathcal{L}_a = \sum_{i,j} \left| \frac{\mathbf{w}_i \cdot \mathbf{w}_j}{\|\mathbf{w}_i\|_2 \|\mathbf{w}_j\|_2} - \tanh\left( k \left( \frac{\mathbf{e}_i \cdot \mathbf{e}_j}{\|\mathbf{e}_i\|_2 \|\mathbf{e}_j\|_2} - \frac{1}{|N|^2} \sum_{k,l} \frac{\mathbf{e}_k \cdot \mathbf{e}_l}{\|\mathbf{e}_k\|_2 \|\mathbf{e}_l\|_2} \right) \right) \right|, \quad (4)$$

where $\mathbf{e}_i$ is the Evo2 embedding for sequence $i$, $\mathbf{w}_i$ is the watermark logit vector produced by the watermark model, $|N|$ is the number of sequences, $k$ is a hyperparameter controlling the similarity range, and $\| \cdot \|_2$ denotes the Euclidean norm. This loss ensures watermark logits reflect DNA functional relationships while enhancing separability.

Following [39], the normalization loss enforces unbiased token preference and balanced scores. It constrains the mean of the watermark logits to zero across tokens and sequences and ensures uniform absolute values for stability. The normalization loss $\mathcal{L}_n$ is defined as:

$$\mathcal{L}_n = \sum_{i=1}^{|N|} \left| \sum_{j=1}^{|\mathcal{V}|} \mathbf{w}_i^{(j)} \right| + \sum_{j=1}^{|\mathcal{V}|} \left| \sum_{i=1}^{|N|} \mathbf{w}_i^{(j)} \right| + \lambda \sum_{i=1}^{|N|} \sum_{j=1}^{|\mathcal{V}|} \left| R - \mathbf{w}_i^{(j)} \right|, \quad (5)$$

where $\mathbf{w}_i^{(j)}$ denotes the $j$-th value in the watermark logit; $R$ is a hyperparameter specifying the target absolute value for each logit component, and $\lambda$ is a weighting factor. This loss ensures the watermark is statistically neutral and detectable. The total loss combines the above two objectives. During watermarked generation, the watermark logits, scaled by a watermark strength factor $\delta$, are added to the original logits to bias the sampling of the next nucleotide.

### 4.1.2 Synonymous Codon Substitutions

To design a function-invariant watermark for DNA language models, DNAMark employs *synonymous codon substitution (SCS)* within the coding DNA sequence (CDS), targeting the *third base* of codons to embed identifiers that preserve the encoded amino acid, critical for synthetic biology applications. For a codon with fixed first two bases (e.g., CA) and an intended amino acid (e.g., Histidine for CAT), DNAMark defines green and red lists within the synonymous codon set (e.g., CAC as red list and CAT as green list), to keep the encoded protein unchanged (i.e., no matter red or green list is chosen, the same amino acid type). This approach is motivated by several considerations: **First**, synonymous codons produce identical amino acids, thereby maintaining the protein's structure and function critical for applications in synthetic biology. **Second**, targeting the third base leverages the degeneracy of the genetic code, where mutations at this position are often silent [31], minimizing the influence of watermarking on DNA sequences. **Third**, by watermarking only the third base, DNAMark achieves a sparse watermark that balances robust detectability with high DNA sequence quality, minimizing disruptions to codon usage and sequence optimality. Following previous works [34, 24], we explicitly define the green and red lists for watermark. Considering different cases of

synonymous codons (more details in Table B), the green and red lists $(\mathcal{G}, \mathcal{R})$ are constructed as:

$$\mathcal{G}, \mathcal{R} = \begin{cases} \{b_g\}, \mathcal{S} \setminus \{b_g\} & \text{if } |\mathcal{S}| = 2 \text{ (e.g., T, C for CAT, CAC; Histidine),} \\ \{b_g\}, \mathcal{S} \setminus \{b_g\} & \text{if } |\mathcal{S}| = 3 \text{ (e.g., T, C, A for ATT, ATC, ATA; Isoleucine),} \\ \{b_g\}, \mathcal{S} \setminus \{b_g\} & \text{if } |\mathcal{S}| = 4 \text{ (e.g., T, C, A, G for GCT/ C/ A/ G; Alanine),} \\ \emptyset, \emptyset & \text{if } |\mathcal{S}| = 1 \text{ (e.g., G for ATG; Methionine),} \end{cases} \quad (6)$$

where $\mathcal{S} = \{b_3 \in \{T, C, A, G\} \mid \texttt{translate}(b_1, b_2, b_3) = a\}$ is the set of third bases yielding the same amino acid $a$, and $|\mathcal{S}|$ is the set size; $b_1, b_2$ are the first two bases, $\texttt{translate}$ maps codons to amino acids; $\{b_g\} \in \mathcal{S}$ is the green base list, selected as the base type with the highest watermark logits in $\mathcal{S}$. For $|\mathcal{S}| = 2$ (e.g., $b_1$=C, $b_2$=A, $a$=Histidine), one base is green (e.g., T for CAT) and one red (e.g., C for CAC); for $|\mathcal{S}| = 3$ (e.g., $b_1$=A, $b_2$=T, $a$=Isoleucine), one is green (e.g., C) and two red (e.g., T, A); for $|\mathcal{S}| = 4$ (e.g., $b_1$=G, $b_2$=C, $a$=Alanine), one is green and three red; and for $|\mathcal{S}| = 1$ (e.g., $b_1$=A, $b_2$=T, $a$=Methionine), watermarking is skipped as no synonymous alternatives exist.

### 4.1.3 Adaptive Watermark Strength and Entropy-guided Watermark

Given the small vocabulary of DNA sequences (A, C, T, G) and the instability of autoregressive DNA language models, where excessive watermarking may produce invalid sequences such as repeated motifs or model corruption, DNAMark employs optimization strategies to balance detectability and sequence quality. Specifically, we introduce two optimization strategies: *Adaptive Watermark Strength* and *Entropy-guided Watermarking*. The **Adaptive Watermark Strength strategy** dynamically adjusts the watermark logit strength, $\delta$, using an Exponential Moving Average (EMA) [29] based on the current z-score, $z_t$, which measures the statistical significance of the watermark signal (i.e., green base frequency in green/red lists [34]). The strength is smoothly updated as a weighted average of the current strength within a target range $[z_{\min}, z_{\max}]$. The adjustment is defined as:

$$\texttt{adj}(z_t, z_{\min}, z_{\max}) = \begin{cases} z_{\min} - z_t & \text{if } z_t < z_{\min}, \\ 0 & \text{if } z_{\min} \leq z_t \leq z_{\max}, \\ z_{\max} - z_t & \text{if } z_t > z_{\max}, \end{cases} \quad (7)$$

and $\delta$ is smoothly updated as a weighted average of the current strength and a target adjustment:

$$\delta_{t+1} = (1 - \beta)\delta_t + \beta \cdot \max\left(\delta_{\min}, \min\left(\delta_{\max}, \delta_t + \kappa \cdot \texttt{adj}(z_t, z_{\min}, z_{\max})\right)\right), \quad (8)$$

where $\delta_t$ is the strength at step $t$, $\beta \in (0, 1)$ controls the update speed, $\delta_{\min}, \delta_{\max}$ are bounds, and $\kappa$ scales the adjustment. If $z_t < z_{\min}$, $\delta$ increases to enhance detectability; if $z_t > z_{\max}$, $\delta$ decreases to preserve sequence quality; and if $z_t \in [z_{\min}, z_{\max}]$, $\delta$ remains stable. During generation, watermark logits, scaled by $\delta_t$, are added to the original logits.

The **Entropy-guided Watermarking strategy** skips watermarking in low-entropy subsequences to avoid disrupting critical sequence patterns, such as regulatory motifs in UTRs. The entropy $H$ of a subsequence $s$ (e.g., a window of nucleotides) is computed as:

$$H(s) = -\sum_{b \in \{T,C,A,G\}} p(b) \log p(b), \quad (9)$$

where $p(b)$ is the frequency of base $b$ in $s$. If $H(s) < H_{\text{threshold}}$, watermarking is skipped for that subsequence, ensuring minimal impact on functional elements like ribosome binding sites or structural motifs. These strategies together enhance DNAMark's watermark, preserving sequence quality while maintaining robust detectability against mutations and adversarial edits.

### 4.2 CentralMark: Inheritable Watermarks from DNA to Proteins

Recent DNA language models not only learns DNA sequences but also captures the central dogma [13]'s flow of genetic information from DNA to RNA to protein [8, 46]. To extend the traceability of our DNA watermark beyond the nucleotide sequence, we introduce an inheritable watermark (CentralMark) *detectable in both generated DNA and the translated protein sequence*, a critical feature to ensure biosecurity and ownership verification in synthetic biology applications where proteins are the functional output (Figure 1 (e)). Unlike DNAMark introduced above, which uses synonymous codon substitutions to preserve protein function, the inheritable watermark deliberately

alters amino acids by targeting the second base of codons in the coding DNA sequence (CDS), leveraging ESM [37] embeddings of the translated protein instead of Evo2 embeddings of DNA for both watermark generation and detection. We target the *second base of each codon* because it predominantly determines the encoded amino acid's identity or chemical properties, facilitating precise amino acid substitutions, and enables near-nonoverlapping green and red lists for amino acids based on second-base patterns (see Table 3). Specifically, for a codon $c = (b_1, b_2, b_3) \in \mathcal{V}_{\text{CDS}}$, where $b_2 \in \{A, C, G, T\}$, we define a green/red list for the protein sequence by indexing the amino acid $a = \texttt{translate}(c)$ to the second base $b_2$:

$$\mathcal{G}_a = \{a \mid \texttt{translate}(b_1, b_2, b_3) = a, b_2 \in \mathcal{G}_b\}, \mathcal{R}_a = \{a \mid \texttt{translate}(b_1, b_2, b_3) = a, b_2 \in \mathcal{R}_b\}, \tag{10}$$

where $\mathcal{G}_b$ and $\mathcal{R}_b$ are the green and red sets of second bases (e.g., $\mathcal{G}_b = \{C, G\}$), and $\texttt{translate}$ maps codons to amino acids (e.g, $\mathcal{G}_a = \{\text{Leu, Pro, His, Gln, Arg, Val, Ala, Asp, Glu, Gly}\}$). During watermarking, we bias the selection of codons with $b_2 \in \mathcal{G}_b$ to embed the signature, which propagates to the protein as a biased distribution of amino acids in $\mathcal{G}_a$. In CentralMark, the green sets of second bases are chosen by selecting the bases with the top-2 highest watermark logits. By embedding watermarks in DNA sequences based on their translated protein sequences, we enable subsequent *detection of the protein sequences independently, without requiring additional DNA information*.

## 4.3 Watermark Detection

The watermark detection of DNAMark and CentralMark follows KGW's calculating z score (Equation. 3). We need to note that the expected proportion of green tokens, $\gamma$, may not be 0.5 in DNAMark and CentralMark due to the unique design, such as synonymous codon substitutions. Under the assumption of uniform codon usage, $\gamma$ is set to 0.3559, 0.5, and 0.55 for DNAMark, CentralMark (DNA), and CentralMark (Protein) respectively. The details are included in the Appendix. E.

## 5 Experiments

### 5.1 Experiment Settings

**BenchMark Construction**   To construct a biologically grounded benchmark for evaluating DNA watermarks, we curated a set of therapeutically important protein-coding genes from *Homo sapiens (Human)* and existing drug modalities. These genes were selected based on their established relevance in clinical and pharmaceutical contexts, encompassing categories such as cytokines (e.g., IL2 [61], TNF [51]), growth factors (e.g., VEGFA [36], EGF [27]), immune checkpoint proteins (e.g., PDCD1 [60], CD274 [19]), apoptosis regulators (e.g., TP53 [41], BCL2 [65]), oncogenes (e.g., KRAS [57], BRAF [17]), antiviral effectors (e.g., IFNA1 [45], TLR3 [3]), coagulation factors (e.g., F8 [64], F2 [68]), and other categories relevant to disease and therapy. For each gene, we queried the NCBI RefSeq database [50] to retrieve validated coding DNA sequences (CDS) with canonical start and stop codons. We integrated secondary structure annotations (helix, $\beta$-strand, loop) from UniProt [66] to ensure structural context. Monomeric proteins with varied secondary structures were selected, constructing a benchmark with 400 DNA sequences (More details in Appendix. C). In Case Study, we explored watermarking CRISPR-Cas9 with both coding and non-coding regions.

**Attacks**   To evaluate the robustness of our proposed watermarking scheme, we subjected the watermarked DNA sequences to a series of simulated genetic alterations, mimicking common evolutionary and mutational processes. These *in silico* attacks comprised three distinct types of modifications: (1) **Synonymous Codon Substitutions** replace codons with alternatives that encode the same amino acid [14, 48] (2) **Nucleotide Substitutions** means changing randomly seleted nucleotides to other types in DNA [54, 67], which can lead to either synonymous or non-synonymous codon changes; and (3) **Insertions and deletions (Indels)**, are structural variants that add or remove nucleotides. Here we consider add or remove codons [43, 44]. These attacks are performed at a frequency of 5% across the sequence to simulate a harsh test for the watermark's detectability and robustness (natural mutation frequency $10^{-3} - 10^{-8}$ [56, 20]).

**Evaluations**   For each DNA sequence, we use the first half as a prompt to the DNA language models and generate the rest for 5 times. Inspired by previous works on LLM watermark [39, 77], we report the detection True positive rates at different false positive rates (1% and 10%) to avoid the

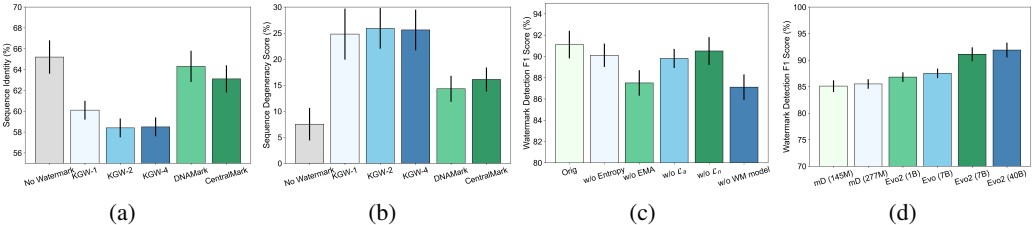

Figure 2: (a) & (b) Generated DNA sequence quality measured by Sequence Identity (the higher the better) and Degeneracy Score (the lower the better). (c) Ablation studies of Entropy Guidance, Adaptive $\delta$ with EMA, Alignment and Normalization loss, and the watermark model. We perform 3-time generations for each model and show the standard deviation. (d) Applying DNAMark to different DNA language models and measuring the watermark detection F1 score. mD: megaDNA.

impact of detection thresholds ($\tau$). To assess the quality of generated DNA sequences, we compute the **Sequence Identity** to the ground truth, where higher values indicate better alignment, and the **Degeneracy Score**, defined as the percentage of a sequence covered by repetitive substrings longer than four nucleotides, where lower values are preferable, following Evo [46].

**DNA Language Models and Baselines** We evaluate DNAMark and CentralMark on the latest and largest DNA language models, Evo [46] and Evo2 (7B, and 40B) [8]. Our methods can also be applied to other DNA models. Hyperparameters are set to $k = 20, \lambda = 10, \kappa = 0.1, \delta_{min} = 0.5, \delta_{max} = 3.5, z_{min} = 2.5, z_{max} = 4.0, H_{threshold} = 2.0$, and the Adam optimizer (lr=1e-3) is used for training (Selected Hyperparamter analysis in Figure 4). We adapt KGW with 1, 2, and 4 codon window sizes to DNA as a baseline. All experiments are conducted on 4 Tesla H100 GPUs.

## 5.2 Results and Robustness Analysis

In Table 1, we compare the performance of DNAMark and CentralMark, detected using DNA and protein sequences, against KGW-1, KGW-2, and KGW-4 under various attack scenarios. Watermarking and detecting DNA sequences is notably more challenging than in natural language models, where methods like KGW achieve near-100% TPR for texts [34], compared to only 70–80% TPR for DNA. We identify two primary reasons for this disparity: (1) DNA's limited vocabulary of four nucleotides (A, C, G, T), versus tens of thousands of tokens in natural language models, severely constrains green/red list assignments, reducing the watermark's statistical distinctiveness. (2) DNA language models exhibit greater brittleness than large language models (LLMs), showing high sensitivity to perturbations in their output distributions. When the watermark strength $\delta$ is excessive, it overly biases nucleotide selection, leading to model collapse (e.g., generating repetitive motifs like AAAAA), which compromises both sequence quality and watermark detectability.

Across all attack conditions, DNAMark and CentralMark consistently outperform KGW baselines in TPR and F1 scores at both 1% and 10% FPR. The detection F1 of DNAMark and CentralMark are all above 0.85. CentralMark (DNA) achieves the highest performance in most cases, followed closely by CentralMark (Protein) and DNAMark. The unique design of CentralMark makes the watermark detectable in both the generated DNA and the translated protein. The robustness of DNAMark and CentralMark is due to their use of embeddings (Evo2, ESM), which capture functional/semantic similarity, making watermarks robust even with attacks. For instance, DNAMark and CentralMark achieve high TPR and F1 scores under synonymous codon substitutions, as these changes preserve amino acid sequences and minimally affect the embeddings.

Comparing different attacks, we observe that Nucleotide Substitutions and Indels are the most strong attacks: Substitutions can lead to non-synonymous codons, and Indels can disrupt sequence patterns critical for watermark integrity. For example, the TRP of CentralMark with Indels drops to around 76%, highlighting the severity of these attacks. Nevertheless, DNAMark and CentralMark outperform all baselines. Future work will focus on enhancing robustness to such challenging attacks.

Table 1: We compared the performance of our watermarking methods, DNAMark and CentralMark (DNA/Protein), with baselines, including KGW-k [34], with DNA language model Evo2-7B [8]. Tests evaluated watermark detection accuracy under no attack, synonymous codon substitution, Nucleotide Substitutions, and insertion-deletion (Indels) attacks.

| Method | No attack | | | | Synonymous Codon Substitution | | | |
| | 1% FPR | | 10% FPR | | 1% FPR | | 10% FPR | |
| | TPR | F1 | TPR | F1 | TPR | F1 | TPR | F1 |
|---|---|---|---|---|---|---|---|---|
| KGW-1 | 0.765 | 0.862 | 0.805 | 0.845 | 0.580 | 0.729 | 0.756 | 0.815 |
| KGW-2 | 0.770 | 0.865 | 0.820 | 0.854 | 0.545 | 0.701 | 0.740 | 0.805 |
| KGW-4 | 0.774 | 0.868 | 0.817 | 0.852 | 0.371 | 0.537 | 0.520 | 0.642 |
| DNAMark | 0.845 | 0.911 | 0.915 | 0.908 | 0.820 | 0.896 | 0.896 | 0.898 |
| CentralMark (DNA) | **0.875** | **0.928** | 0.920 | 0.911 | 0.854 | 0.916 | **0.910** | **0.905** |
| CentralMark (Protein) | 0.868 | 0.924 | **0.922** | **0.912** | **0.860** | **0.920** | 0.904 | 0.902 |

| Method | Nucleotide Substitutions | | | | Indels | | | |
| | 1% FPR | | 10% FPR | | 1% FPR | | 10% FPR | |
| | TPR | F1 | TPR | F1 | TPR | F1 | TPR | F1 |
|---|---|---|---|---|---|---|---|---|
| KGW-1 | 0.520 | 0.680 | 0.710 | 0.785 | 0.515 | 0.675 | 0.723 | 0.794 |
| KGW-2 | 0.505 | 0.667 | 0.658 | 0.749 | 0.477 | 0.642 | 0.645 | 0.739 |
| KGW-4 | 0.330 | 0.493 | 0.551 | 0.668 | 0.339 | 0.503 | 0.497 | 0.623 |
| DNAMark | 0.808 | 0.902 | 0.886 | 0.892 | **0.795** | **0.878** | **0.860** | **0.877** |
| CentralMark (DNA) | **0.840** | **0.908** | **0.890** | **0.894** | 0.765 | 0.862 | 0.850 | 0.872 |
| CentralMark (Protein) | 0.825 | 0.900 | 0.885 | 0.892 | 0.759 | 0.858 | 0.832 | 0.861 |

## 5.3 Generation Quality and Ablation Studies

It is important to keep the sequence quality when watermarking DNA for practical use. In Figure 2 (a) & (b), we show the Sequence Identity to the ground truth and the Degeneracy Score of the generated DNA sequences by different watermark methods. Compared with KGW, DNAMark and CentralMark shows more alignment with no watermark, indicating higher generation quality. This can be attributed to the sparse watermark adapted to DNA and unique methods such as synonymous codon substitution of DNAMark, minimizing the side-effects on sequence quality. In Figure 2 (c), we did ablation studies of various components in DNAMark. Generally, Adaptive watermark strength with EMA and the watermark model are most critical to the successful watermark detection.

## 5.4 Generalization to Different DNA Models and Time Complexity

In Figure 2 (d), we observe that DNAMark demonstrates robust watermark detection across a range of DNA models. Using models of varying sizes—megaDNA (145M and 277M parameters), Evo2 (1B, 7B, and 40B), and Evo1 (7B)—DNAMark achieves F1 scores from 0.851 to 0.919. Smaller models, such as megaDNA-145M (F1=0.851) and 277M (F1=0.855), deliver respectable detection accuracy, but are limited by reduced generation capability. Larger models like Evo2-7B (F1=0.911) and Evo2-40B (F1=0.919) excel, leveraging high-capacity embeddings to enhance generation quality and watermark detection. We further measure the generation time cost of DNAMark and CentralMark, comparing them to a baseline with no watermark generation. The time complexity increases by approximately 30% (Table 7), attributable to the compact size of the watermark model.

## 5.5 Case Study of Watermarking CRISPR-Cas9 System

To show the practical application in gene editing, we utilized the Evo model (evo-1-8k-crispr) to generate the CRISPR-Cas9 [11, 12] DNA sequences, embedding a watermark during generation using CentralMark. Following [46], we use Prodigal [32] to extract Cas9 CDS, MinCED [6] to detect CRISPR arrays, and AlphaFold3 (AF3) [2] to predict the structure. Figure 3 visualizes the generated watermarked Cas9 aligned with the wild-type SpCas9 crystal structure (PDB ID: 4OO8). The generated sequence achieves a TM-score of 0.6802, indicating high structural alignment, and a

Z-score of 5.41, confirming strong watermark detectability. These results demonstrate the efficacy of watermarking Evo-generated CRISPR-Cas9 DNA sequences with minimal impact on biological quality.

## 6 Conclusions

In this paper, we tackle the pressing biosecurity challenges arising from DNA language models, which hold immense potential for genetic engineering but also pose dual-use risks by enabling the creation of harmful biological agents. To counter these risks, we propose DNAMark, a watermarking method that uses synonymous codon substitutions to embed robust, function-preserving watermarks in DNA sequences, and CentralMark, an advanced technique that generates inheritable watermarks detectable in both DNA and translated proteins. Future work should explore watermark schemes independent of green/red lists to enhance adaptability, investigate their effects on UTRs for regulatory insights, and validate DNAMark and CentralMark through wet lab experiments. These steps are vital to responsibly balance genetic technology innovation with biosecurity.

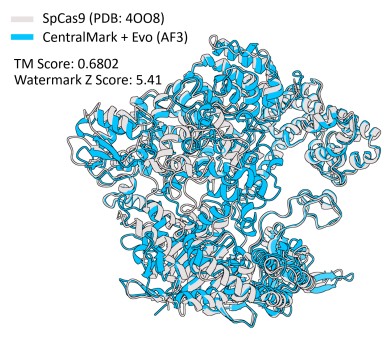

Figure 3: Predicted structure of Evo-desinged Cas9 with CentralMark.

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

# A   Broad Impacts

The societal implications of DNAMark and CentralMark are profound and multifaceted. On the positive side, these watermarking techniques mitigate biosecurity threats by enabling researchers, regulators, and biosafety organizations to track and verify the origins of synthetic DNA, deterring malicious applications such as the engineering of pathogens. This traceability fosters trust in synthetic biology, supporting advancements in therapeutics, agriculture, and environmental solutions. Moreover, by establishing a framework for responsible innovation, these methods could encourage international collaboration on biosecurity standards, strengthening global oversight of genetic technologies. However, negative consequences must also be considered. The watermarking methods may not be entirely impervious to circumvention by sophisticated adversaries who could exploit vulnerabilities, such as reverse-engineering watermarks or introducing mutations to obscure them. This limitation risks fostering a false sense of security among stakeholders, potentially undermining trust in regulatory frameworks if breaches occur. Additionally, the computational and expertise barriers to implementing these watermarks could disproportionately burden smaller research institutions or developing nations, exacerbating inequities in access to cutting-edge genetic technologies. In the future, we will further refine our watermark methods and establish a community to advance watermarking research and reduce the potential negative impacts.

# B   Codon-to-Amino-Acid Table

| 1st/2nd | U | C | A | G |
|---|---|---|---|---|
| **U** | UUU Phe
UUC Phe
UUA Leu
UUG Leu | UCU Ser
UCC Ser
UCA Ser
UCG Ser | UAU Tyr
UAC Tyr
UAA Stop
UAG Stop | UGU Cys
UGC Cys
UGA Stop
UGG Trp |
| **C** | CUU Leu
CUC Leu
CUA Leu
CUG Leu | CCU Pro
CCC Pro
CCA Pro
CCG Pro | CAU His
CAC His
CAA Gln
CAG Gln | CGU Arg
CGC Arg
CGA Arg
CGG Arg |
| **A** | AUU Ile
AUC Ile
AUA Ile
AUG Met | ACU Thr
ACC Thr
ACA Thr
ACG Thr | AAU Asn
AAC Asn
AAA Lys
AAG Lys | AGU Ser
AGC Ser
AGA Arg
AGG Arg |
| **G** | GUU Val
GUC Val
GUA Val
GUG Val | GCU Ala
GCC Ala
GCA Ala
GCG Ala | GAU Asp
GAC Asp
GAA Glu
GAG Glu | GGU Gly
GGC Gly
GGA Gly
GGG Gly |

Table 2: Standard RNA codon table organized by the first two nucleotides. Each cell shows four codons sharing the same first two bases.

| Second Base | Amino Acids |
|---|---|
| A | Isoleucine (Ile), Methionine (Met), Threonine (Thr), Asparagine (Asn), Lysine (Lys), Serine (Ser), Arginine (Arg) |
| C | Leucine (Leu), Proline (Pro), Histidine (His), Glutamine (Gln), Arginine (Arg) |
| G | Valine (Val), Alanine (Ala), Aspartic Acid (Asp), Glutamic Acid (Glu), Glycine (Gly) |
| T | Phenylalanine (Phe), Leucine (Leu), Serine (Ser), Tyrosine (Tyr), Cysteine (Cys), Tryptophan (Trp), Stop |

Table 3: Second base to amino acid mapping for the standard genetic code. This table lists the amino acids corresponding to each possible second base (A, C, G, T) in codons of the coding DNA sequence (CDS), used for CentralMark's inheritable watermark, where the second base is modified to embed a detectable signature in the translated protein.

| 3-Letter | 1-Letter | 3-Letter | 1-Letter | 3-Letter | 1-Letter |
|---|---|---|---|---|---|
| Ala | A | Gly | G | Pro | P |
| Arg | R | His | H | Ser | S |
| Asn | N | Ile | I | Thr | T |
| Asp | D | Leu | L | Trp | W |
| Cys | C | Lys | K | Tyr | Y |
| Glu | E | Met | M | Val | V |
| Gln | Q | Phe | F | | |

Table 4: Amino Acid Three-Letter to One-Letter Code Mapping

# C  Therapeutic DNA Benchmark

Table 5: Statistics of CDS sequences in each therapeutic category.

| Category | Count | Avg Length | Min Length | Max Length |
|---|---|---|---|---|
| Cytokines | 15 | 556.20 | 282 | 759 |
| GrowthFactors | 77 | 665.88 | 180 | 3501 |
| ImmuneCheckpoints | 14 | 816.21 | 525 | 1578 |
| Hormones | 18 | 431.67 | 333 | 654 |
| Apoptosis | 58 | 848.90 | 471 | 1182 |
| Enzymes | 31 | 4101.10 | 912 | 7650 |
| Coagulation | 9 | 2630.33 | 651 | 7056 |
| Transporters | 7 | 3706.71 | 1479 | 4443 |
| Oncogenes | 31 | 1735.35 | 567 | 2424 |
| Antiviral | 3 | 2000.00 | 570 | 2715 |
| Neurotrophins | 28 | 1052.68 | 726 | 2391 |
| Uncategorized | 112 | 1784.22 | 255 | 5028 |

Table 6: Representative therapeutic genes by category.

| Category | Genes |
|---|---|
| Cytokines | IL2, IL6, IL10, TNF, IFNG |
| GrowthFactors | EGF, FGF1, VEGFA, PDGFA, TGFB1 |
| ImmuneCheckpoints | PDCD1, CD274, CTLA4, LAG3 |
| Hormones | INS, LEP, GH1, PTH |
| Apoptosis | BCL2, CASP3, TP53 |
| Enzymes | JAK1, CDK4, MAPK1, MTOR |
| Coagulation | F8, F9, F2 |
| Transporters | ABCB1, CFTR, SLC2A1 |
| Oncogenes | KRAS, BRAF, MYC |
| Antiviral | IFNA1, IFNB1, TLR3 |
| Neurotrophins | NGF, BDNF, NTRK1 |

# D  More Results of DNAMark and CentralMark

| Watermarking Method | Evo(7B) | Evo(40B) |
|---|---|---|
| No Watermark | 9.7 | 30.5 |
| DNAMark | 12.4 | 37.2 |
| CentralMark | 13.2 | 40.5 |

Table 7: Generation times (in seconds) for producing a 128-nucleotide DNA sequence using No Watermark, DNAMark, and CentralMark on Evo(7B) and Evo(40B) models. DNAMark and Central-Mark incur computational overhead due to obtaining embedding and watermark model computations.

# E Calculation Details of $\gamma$

## E.1 Calculation of Expected Green Token Proportion ($\gamma$) for DNAMark

In the DNAMark watermarking scheme, $\gamma$ represents the expected proportion of green tokens (third bases in the green list $\mathcal{G}$) under the null hypothesis of no watermark, where the first two bases of codons are uniformly distributed. This calculation is performed for watermarkable positions, i.e., codons with synonymous third bases $|\mathcal{S}| \geq 2$, as defined in Equation (6) and detailed in Appendix B. The process is summarized as follows:

1. **Identify watermarkable codons**: For each codon prefix $(b_1, b_2)$, uniformly distributed over 16 possibilities (probability $\frac{1}{16}$), the synonymous set $\mathcal{S} = \{b_3 \in \{T, C, A, G\} \mid \texttt{translate}(b_1, b_2, b_3) = a\}$ determines the number of third bases encoding the intended amino acid $a$. Excluding stop codons, the 61 sense codons yield 59 watermarkable codons: 32 with $|\mathcal{S}| = 4$ (e.g., Alanine: GCT, GCC, GCA, GCG), 3 with $|\mathcal{S}| = 3$ (e.g., Isoleucine: ATT, ATC, ATA), and 24 with $|\mathcal{S}| = 2$ (e.g., Histidine: CAT, CAC).

2. **Assign green list probability**: For each watermarkable codon, the green list $\mathcal{G} = \{b_g\}$ contains one base from $\mathcal{S}$, selected as the base with the highest watermark logits. Under the null hypothesis, the third base is chosen uniformly from $\mathcal{S}$, so the probability of selecting the green base is $\frac{1}{|\mathcal{S}|}$.

3. **Compute $\gamma$**: The expected proportion $\gamma$ is the weighted average of $\frac{1}{|\mathcal{S}|}$ over all watermarkable codons, weighted by their counts:

$$\gamma = \frac{\sum_{k=2}^{4}(\text{number of codons with } |\mathcal{S}| = k) \times \frac{1}{k}}{\text{total watermarkable codons}}.$$

Calculating contributions: $32 \times \frac{1}{4} = 8$ for $|\mathcal{S}| = 4$, $3 \times \frac{1}{3} = 1$ for $|\mathcal{S}| = 3$, and $24 \times \frac{1}{2} = 12$ for $|\mathcal{S}| = 2$. Total $= 8 + 1 + 12 = 21$. With 59 watermarkable codons, $\gamma = \frac{21}{59} \approx 0.3559$.

This $\gamma$ value serves as the baseline for watermark detection, enabling the z-score calculation to identify the presence of a watermark by comparing observed green base frequencies against this expected proportion.

## E.2 Calculation of Expected Green Amino Acid Proportion ($\gamma$) for CentralMark

In the CentralMark watermarking scheme, $\gamma$ represents the expected proportion of green amino acids in the translated protein sequence under the null hypothesis, assuming a uniform distribution over the 20 standard amino acids. The watermark targets the second base of codons, with the green set $\mathcal{G}_b$ comprising the two bases with the top-2 watermark logits from $\{A, C, G, T\}$, as defined in Equation (10) and detailed in Table 3. Since $\mathcal{G}_b$ is not fixed, we average $\gamma$ over all possible pairs $\mathcal{G}_b \in \{\{A, C\}, \{A, G\}, \{A, T\}, \{C, G\}, \{C, T\}, \{G, T\}\}$. The process is summarized as follows:

1. **Identify green amino acids**: For each $\mathcal{G}_b$, the green amino acids $\mathcal{G}_a = \{a \mid \texttt{translate}(b_1, b_2, b_3) = a, b_2 \in \mathcal{G}_b\}$ are the union of amino acids associated with the two second bases, per Table 3 (e.g., $\mathcal{G}_b = \{A, C\}$ yields 11 amino acids).

2. **Compute per-pair $\gamma$**: For each $\mathcal{G}_b$, $\gamma = \frac{\text{Number of unique amino acids in } \mathcal{G}_a}{20}$, reflecting the uniform probability $\frac{1}{20}$ per amino acid. Values range from $\frac{10}{20}$ (e.g., $\{C, G\}$) to $\frac{12}{20}$ (e.g., $\{A, G\}$).

3. **Average $\gamma$**: Assuming equal likelihood for each $\mathcal{G}_b$, the average is:

$$\gamma = \frac{1}{6}\left(\frac{11}{20} + \frac{12}{20} + \frac{12}{20} + \frac{10}{20} + \frac{10}{20} + \frac{11}{20}\right) = \frac{11}{20} = 0.55.$$

This $\gamma$ serves as the baseline for detecting the CentralMark watermark in protein sequences, enabling z-score calculations to identify biased amino acid distributions.

## F  Influence of Hyperparameter Selection

In Figure. 4, we show the influence of $\delta_{max}$ on the watermark detection F1 and sequence degeneracy score of DNAMark. We observe that too large $\delta_{max}$ may lead to worse sequence quality measured by degeneracy, and $\delta_{max}$ in a suitable range maximizes detection F1. In experiments, we choose $\delta_{max} = 3.5$ as the default setting.

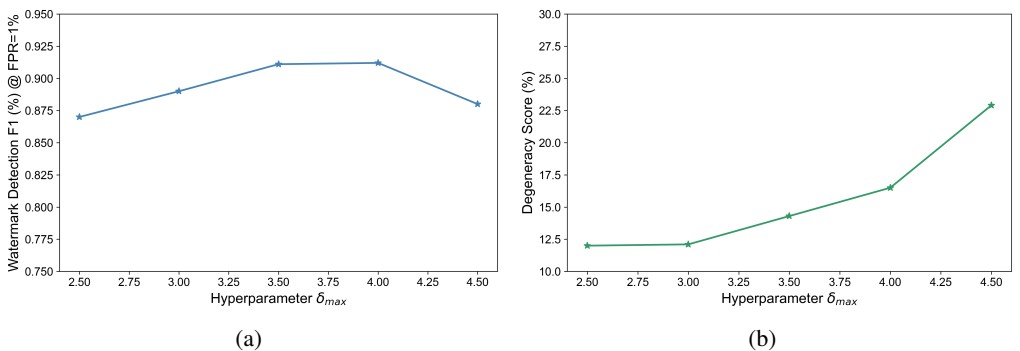

(a)                           (b)

Figure 4: Hyperparamter analysis of $\delta_{max}$

## G  Details of Watermark Model

Our watermark model adopts an architecture similar to SIR [39], consisting of a series of residual blocks with ReLU activation, as detailed in the code. However, our implementation incorporates additional LayerNorm layers after each residual block to stabilize training and improve convergence. Notably, the input embeddings for our model are derived from Evo2 (7B) and ESM2 (35M), leveraging their robust representations to enhance the model's ability to capture the biological semantics of DNA/protein sequences. To train the watermark model, we crawl 1000 random human coding sequences (CDS) from RefGen, subsample them to extract 20-length codons/amino acid embeddings with the Evo/Evo2 and ESM2 as input, and fine-tune the model for 200 epochs using the combination of alignment and normalization loss (Equation. 4 and 5). More details of code are included at `https://anonymous.4open.science/r/DNA_Watermark-1687/README.md` .

```python
class ResidualBlock(nn.Module):                                             1
    def __init__(self, dim):                                                2
        super(ResidualBlock, self).__init__()                               3
        self.fc = nn.Linear(dim, dim)                                       4
        self.relu = nn.ReLU()                                               5
                                                                            6
    def forward(self, x):                                                   7
        out = self.fc(x)                                                    8
        out = self.relu(out)                                                9
        out = out + x                                                       10
        return out                                                          11
                                                                            12
class WatermarkModel(nn.Module):                                            13
    def __init__(self, num_layers=4, input_dim=1024, hidden_dim=500,        14
        output_dim=4):
        super(TransformModel, self).__init__()                              15
        self.layers = nn.ModuleList()                                       16
        self.norms = nn.ModuleList()                                        17
        self.layers.append(nn.Linear(input_dim, hidden_dim))                18
        self.norms.append(nn.LayerNorm(hidden_dim))                         19
        for _ in range(num_layers - 2):                                     20
            self.layers.append(ResidualBlock(hidden_dim))                   21
            self.norms.append(nn.LayerNorm(hidden_dim))                     22
        self.layers.append(nn.Linear(hidden_dim, output_dim))              23
        self.norms.append(nn.LayerNorm(output_dim))                         24
                                                                            25
    def forward(self, x):                                                   26
        for i in range(len(self.layers)):                                   27
            x = self.layers[i](x)                                           28
            x = self.norms[i](x)                                            29
        return x                                                            30
```

# H Case Study of CRISPR-Cas9 Design with CentralMark

Here, we show the designed Cas9 sequence with CentralMark + Evo, aligned with the wild type. The total DNA similarity is 67.3%.

```
CentralMark:  1     M N K P Y S I G L D I G T N S V G W S I I T D D Y K V P A K K M R V L G N T D K E Y I K K N L I G A L L F D G G N T A A D R R L K R T A R R   70
                    | : | . | | | | | | | | | | | | | : : | | | | | | | | : | | . : | | | | | | : . . | | | | | | | | | | | | . | . | | . . | | | | | | | | |
Cas9 Ref  :         M D K K Y S I G L D I G T N S V G W A V I T D D Y K V P S K K F K V L G N T D R H S I K K N L I G A L L F D S G E T A E A T R L K R T A R R

CentralMark:  71    R Y T R R R N R I L Y L Q E I F A E E M S K V D D S F F H R L E D S F L V E E D K R G S K Y P I F A T L Q E E K D Y H E K F S T I Y H L R K   140
                    | | | | | : | | | . | | | | | | : . | | : | | | | | | | | | | | | : | | | | | | | | : . . . : | | | . . : . : : | . . | | | | | : . | | | | | | | |
Cas9 Ref  :         R Y T R R K N R I C Y L Q E I F S N E M A K V D D S F F H R L E E S F L V E E D K K H E R H P I F G N I V D E V A Y H E K Y P T I Y H L R K

CentralMark:  141   E L A D K K E K A D L R L I Y I A L A H I I K F R G H F L I E D D S F D V R N T D I S K Q Y Q D F L E I F N T T F E N N D L L S Q N V D V E   210
                    : | | | . . : | | | | | | | | | : | | | | : | | | | | | | | | | . |   . : . . | : | : . | . : . . . : : . : | . . | | . | . : . : . . | | . :
Cas9 Ref  :         K L A D S T D K A D L R L I Y L A L A H M I K F R G H F L I E G D - L N P D N S D V D K L F I Q L V Q T Y N Q L F E E N P I N A S R V D A K

CentralMark:  211   A I L T D K I S K S A K K D R I L A Q Y P N Q K S T G I F A E F L K L I V G N Q A D F K K Y F N L E D K T P L Q F A K D S Y D E D L E N L L   280
                    | | | : . : : | | | . : . : . : : | | . | . : | . . | : | . . . : . | : : | . . . : | | . . | : | . : . . . | | . : | | : | | : | | : | | |
Cas9 Ref  :         A I L S A R L S K S R R L E N L I A Q L P G E K K N G L F G N L I A L L L G L T P N F K S N F D L A E D A K L Q L S K D T Y D D D L D N L L

CentralMark:  281   G Q I G D E F A D L F S A A K K L Y D S V L L S G I L T V I D L S T K A P L S A S M I Q R Y D E H R E D L K Q L K Q F V K A S L P E K Y Q E   350
                    . | | | | : : | | | | . | | | . | . | : : | | | . | | . | . . . | | | | | | | | | | : | | | | | . : | | . . | | . . | : . . | | | | | : |
Cas9 Ref  :         A Q I G D Q Y A D L F L A A K N L S D A I L L S D I L R V N S E I T K A P L S A S M I K R Y D E H H Q D L T L L K A L V R Q Q L P E K Y K E

CentralMark:  351   I F A D S S K D G Y A G Y I E G K T N Q E A F Y K Y L S K L L T K Q E D S E N F L E K I K N E D F L R K Q R T F D N G S I P H Q V H L T E L   420
                    | | . | . | | : | | | | | | : | . . : | | . | | | : : . . : | . | . : . : | . . | . | : . . | | . | | | | | | | | | | | | | | | : | | . | |
Cas9 Ref  :         I F F D Q S K N G Y A G Y I D G G A S Q E E F Y K F I K P I L E K M D G T E E L L A K L N R E D L L R K Q R T F D N G S I P H Q I H L G E L

CentralMark:  421   K A I I R R Q S E Y Y P F L K E N Q D R I E K I L T F R I P Y Y I G P L A R E K S D F A W M T R K T D D S I R P W N F E D L V D K E K S A E   490
                    . | | : | | | . : : | | | | | : | : : | | | | | | | | | | | | | | : | | | | | . . | . | | | | | | | : : : : | . | | | | | : : | | | . . | | :
Cas9 Ref  :         H A I L R R Q E D F Y P F L K D N R E K I E K I L T F R I P Y Y V G P L A R G N S R F A W M T R K S E E T I T P W N F E E V V D K G A S A Q
```

Figure 5: Aligning CentralMark + Evo designed Cas9 to the wild type Cas9 protein sequence.

```
CentralMark: 491    A F I H R M T N N D F Y L P E E K V L P K H S L I Y E K F T V Y N E L T K V R Y K N E - Q G E T Y F F D S N I K Q E I F D G V F K E H R K V   560
                    : | | . | | | | . | . . | | . | | | | | | | | | : | | . | | | | | | | | | : | . . |   . . : . . | . . . . . | : . | . | . : | | . : | | |
Cas9 Ref   :        S F I E R M T N F D K N L P N E K V L P K H S L L Y E Y F T V Y N E L T K V K Y V T E G M R K P A F L S G E Q K K A I V D L L F K T N R K V

CentralMark: 561    S K K K L L D F L A K E Y E E F R I V D V I G L D K E N K A F N A S L G T Y H D L E K I L - D K D F L D N P D N E S I L E D I V Q T L T L F   630
                    : . | : | . : . . . | : . | . | . . | : : . | : : . .       | | | | | | | | | | . | | :   | | | | | | | . : | | . | | | | | | . | | | | |
Cas9 Ref   :        T V K Q L K E D Y F K K I E C F D S V E I S G V E D R - - - F N A S L G T Y H D L L K I I K D K D F L D N E E N E D I L E D I V L T L T L F

CentralMark: 631    E D R E M I K K R L E N Y K D L F T E S Q L K K L Y R R H Y T G W G R L S A K L I N G I R D K E S Q K T I L D Y L I D D G R S N R N F M Q L   700
                    | | : | | | : : | | : . | . . | | . : . . : | : | . | | . | | | | | | | | . | | | | | | | | | : | . | | | | | : | . . | | . : | | | | | | |
Cas9 Ref   :        E D K E M I E E R L K K Y A H L F D D K V M K Q L K R R R Y T G W G R L S R K L I N G I R D K Q S G K T I L D F L K S D G F A N R N F M Q L

CentralMark: 701    I N D D G L S F K S I I S K A Q A G S H S D N L K E V V G E L A G S P A I K K G I L Q S L K I V D E L V K V M G - Y E P E Q I V V E M A R E   770
                    | : | | . | : | | . . | . | | | . . . . | : | . | . : . . | | | | | | | | | | | | | | | : : | : | | | | | | | | |   : : | | . | | : | | | |
Cas9 Ref   :        I H D D S L T F K E D I Q K A Q V S G Q G D S L H E H I A N L A G S P A I K K G I L Q T V K V V D E L V K V M G R H K P E N I V I E M A R E

CentralMark: 771    N Q T T N Q G R R N S R Q R Y K L L D D G V K N L A S D L N G N I L K E Y P T D N Q A L Q N E R L F L Y Y L Q N G R D M Y T G E A L D I D N   840
                    | | | | . : | : : | | | : | . | . : : : | : | . | . | |       | | | | | | | . : | . . | | | | : | : | | | | | | | | | | | | | . . : . | | | : .
Cas9 Ref   :        N Q T T Q K G Q K N S R E R M K R I E E G I K E L G S D - - - - I L K E Y P V E N T Q L Q N E K L Y L Y Y L Q N G R D M Y V D Q E L D I N R

CentralMark: 841    L S Q Y D I D H I I P Q A F I K D D S I D N R V L V S S A K N R G K S D D V P S L E I V K D C K V F W K K L L D A K L M S Q R K Y D N L T K   910
                    | | . | | : | | | : | | : | : | | | | | | | | : | | . . | . | | | | | | | : | | | . | : | | . . | . : | | : | | : | | | : : | | | : | | | | |
Cas9 Ref   :        L S D Y D V D H I V P Q S F L K D D S I D N K V L T R S D K N R G K S D N V P S E E V V K K M K N Y W K Q L L N A K L I T Q R K F D N L T K

CentralMark: 911    A E R G G L T S D D K A R F I Q R Q L V E T R Q I T K H V A R I L D E R F N N E L D S K G R R I R K V - I V T L K S N L V S N F R K E F G F   980
                    | | | | | | : . . | | | . | | : | | | | | | | | | | | | | | | : | | | . | . | . : . | . . : . | | : |   : : | | | | . | | | : | | | : | . |
Cas9 Ref   :        A E R G G L S E L D K A G F I K R Q L V E T R Q I T K H V A Q I L D S R M N T K Y D E N D K L I R E V R V I T L K S K L V S D F R K D F Q F
```

Figure 6: Aligning CentralMark + Evo designed Cas9 to the wild type Cas9 protein sequence.

```
CentralMark: 981   Y K I R E V N N Y H H A H D A Y L N A V V A K A I L T K Y P Q L E P E F V Y G D Y P K Y N S Y K T - R K S - - - - - - A T E K L F F Y S N I   1050
                   | | : | | : | | | | | | | | | | | | | | | | . . | : : . | | | : | | . | | | | | | | . . | : . . | . . | |         | | . | . | | | | | |
Cas9 Ref   :       Y K V R E I N N Y H H A H D A Y L N A V V G T A L I K K Y P K L E S E F V Y G D Y K V Y D V R K M I A K S E Q E I G K A T A K Y F F Y S N I

CentralMark: 1051  M N F F K T K V T L A D G T V V V K D D I E V N N D T G E I V W D K K K H F A T V R K V L S Y P Q N N I V K K T E I Q T G G F S K E S I L A   1120
                   | | | | | | : : | | | : | . : . . : . . | | . | . | . : | | | | | | | | . : . | | | | | | | | | . | | . | | | | | | | : | | | | | | | | | | .
Cas9 Ref   :       M N F F K T E I T L A N G E I R K R P L I E T N G E T G E I V W D K G R D F A T V R K V L S M P Q V N I V K K T E V Q T G G F S K E S I L P

CentralMark: 1121  H G N S D K L I P R K T K D I Y L D P K K Y G G F D S P I V A - Y S V L V V A D I K K G K A Q L K T V T E L L G I T I M E R S R F E K N P   1190
                   . . | | | | | | . | | . | |     . | | | | | | | | | | | . | |   | | | | | | | . : : | | | : : | | | : | . | | | | | | | | | | | | . | | | | |
Cas9 Ref   :       K R N S D K L I A R K - K D - - W D P K K Y G G F D S P T V A L Y S V L V V A K V E K G K S K K L K S V K E L L G I T I M E R S S F E K N P

CentralMark: 1191  S A F L E S K G Y L N I R A D K L I I L P K Y S L F E L E N G R R R L L A S A G E L Q K G N E L A L P T Q F M K F L Y L A S R Y N E S K G K   1260
                   . . | | | : | | | . . : | . | . : | . | | | | | | | | | | | | | | | | : | : | | | | | | | | | | | | | | | | | : : : : . | | | | | | . | . : . | | .
Cas9 Ref   :       I D F L E A K G Y K E V R K D L I I K L P K Y S L F E L E N G R K R M L A S A G E L Q K G N E L A L P S K Y V N F L Y L A S H Y E K L K G S

CentralMark: 1261  P E E I E K K Q E F V N Q H V S Y F D D I L Q L I N D F S K R V I L A D A N L E K I N K L Y Q D N K E N I S V D E L A N N I I N L F T F T S   1330
                   | | : . | : | | . | | . | | . . | . | : | : : . | : : | | | | | | | | | | | | | | | | : | : . . . | . . : : : . . : . | . | . | | | : | | | . | :
Cas9 Ref   :       P E D N E Q K Q L F V E Q H K H Y L D E I I E Q I S E F S K R V I L A D A N L D K V L S A Y N K H R D K - P I R E Q A E N I I H L F T L T N

CentralMark: 1331  L G A P A A - F K F F D K I V D R K R Y T S T K E V L N S T L I H Q S I T G L Y E T R I D L G K L G E D   1382
                   | | | | | |   | | : | | . . : | | | | | | | | | | | | | : : | | | | | | | | | | | | | | | | | . : | | . |
Cas9 Ref   :       L G A P A A A F K Y F D T T I D R K R Y T S T K E V L D A T L I H Q S I T G L Y E T R I D L S Q L G G D
```

Figure 7: Aligning CentralMark + Evo designed Cas9 to the wild type Cas9 protein sequence.

