# OpenReview forum: "Securing the Language of Life: Inheritable Watermarks from DNA Language Models to Proteins"
_NeurIPS.cc/2025/Conference — NeurIPS 2025 poster_

### Official Review · Reviewer_P59q · 2025-07-02

**Clarity:** 3
**Significance:** 3
**Originality:** 2
**Rating:** 4
**Confidence:** 4

**Summary:**

This paper addresses biosecurity concerns arising from the increasing capabilities of DNA language models by introducing two watermarking techniques, DNAMark and CentralMark. DNAMark embeds function-preserving watermarks within synthetic DNA sequences using synonymous codon substitutions, while CentralMark expands this approach by embedding inheritable watermarks detectable in both DNA and the resulting protein sequences through protein embedding-based generation strategies. Both approaches leverage semantic embeddings (Evo2 or ESM) to generate robust watermarks, aiming to balance detection reliability, functional preservation, and robustness to synthesis errors or adversarial edits. The methods are evaluated on a therapeutic DNA benchmark, demonstrating F1 detection scores above 0.85 under a range of mutations and attacks, and maintain a high level of biological sequence quality. A case study on watermarked CRISPR-Cas9 design further illustrates practical applicability.

**Questions:**

1. In Fig 2(c), how could 3 samples yield such small F1 score standard deviations? Could you please confirm these are calculated from only 3-time generations.

2. Why only second and third base are changed for watermarking? Why not first base?

**Ethical Concerns:**

["NO or VERY MINOR ethics concerns only"]

**Final Justification:**

Most of my concerns have been addressed. Please add appropriate citations in the methodology section to acknowledge the contributions of prior work.

**Limitations:**

- Lack technical innovation in watermarking methods

- Non-distortion-free watermarking that degrades output quality and affects biological function

- Limited evaluation of structural and functional impacts

**Paper Formatting Concerns:**

There is no formatting issues。

**Quality:**

2

**Strengths And Weaknesses:**

Strength:

1. The authors propose two dedicated watermarking schemes for DNA LMs: DNAMark, which preserves function via synonymous codon substitutions, and CentralMark, which makes the watermark inheritable and detectable at the protein level by using semantic embeddings. These approaches are adapted logically to the biological constraints of DNA/protein synthesis and expand watermarking beyond the standard green/red list approaches.

2. The experimental results underscore the greater difficulty of DNA watermarking compared with text LLMs, and convincingly show that the proposed methods increase robustness over standard approaches (see, e.g., Table 1).


Weaknesses:

1. Key references of DNA watermarking and its history are missing in this paper. The authors should include an additional section of related works of DNA watermarking. For example, *DNA watermarks: A proof of concept* by D Heider and A Barnekow. This is just an example, many previous works on synthetic biology have the similar idea. The authors should do a detailed related works search and acknowledge previous pioneering works.

2. DNA watermarking concept is not new in biology. Usually, due to biology, the DNA watermark is added on the third base of a codon. See the suggested reference above. Therefore, the claimed innovation point 2 does not hold.

3. The watermarking strategy does not fully use the structure of the genetic code. For DNAMark, the method is constrained to altering only the third base. However, for several amino acids (e.g., Serine: TCT/TCC/TCA/TCG vs AGT/AGC, also Leucine, Arginine), synonymous codons exist where the first or second bases also differ. The paper provides no justification for ignoring these possibilities, which could offer a richer space for watermark embedding. For CentralMark, this method alters the second base to change the resulting amino acid. It is unclear why the first base, which also decisively changes the amino acid, was excluded from consideration. This arbitrary choice limits the scope of the method without clear reasoning.



4. Lack of Technical Novelty: The watermarking methods in this paper, like SIR, KGW are existing method, but adapted to DNA sequence.

5. The authors misuse SIR methodology.

- They incorrectly claim "unbiased distribution" is the second property of SIR (line 161), but the original SIR paper never mentions this. The SIR paper discusses "unbiased token preference," which is a different concept than the unbiasedness in the sense defined by Hu et al. [29]. The method used in this paper, which adds a bias into the logits, is fundamentally a distortion-based watermark and will degrade DNA sequence function.

- The SIR method has a complex training loss to preserve the semantic-consistent broad range property. This paper also uses the same loss, claiming that "logits should be varied sufficiently". However, this paper then only select the maximum watermark logit value, and discard the continuous watermark logits that is produced by the watermark model, abandoning the core properties of SIR method.

- In language model watermarking, there is a series of distortion-free watermarks [1-4] that provably preserve the original models distribution. Those algorithms can be naturally adapted to the DNA LMs. As the proposed watermarking methods cannot provide theoretical guarantee on the distribution bias, it could be outperformed by the distortion-free watermarks adapted to DNA models.

6. Evaluation Concerns:

There are two major measures of watermarking: how does it affect the quality of the output, how detectable it is.

For the quality part:

- Insufficient evidence to support the paper's claims of preserving function. The evaluation of protein structure and function is limited to a single case study on CRISPR-Cas9. Despite having many proteins in their benchmark, the authors didn't study structure and function of other sequences.
- Unclear functional metrics: While TM-Score measures global structural similarity between proteins, its connection to protein function is unclear. For instance, can a TM-score of 0.6802 serve as evidence that watermarked CRISPR-Cas9 retains gene-editing capability? The authors need to clarify this critical point.
- The reported increased degeneracy scores and Sequence Identity shows that watermarking leads to quality degeneration. Recent watermarking literature has developed distortion-free methods, but this work doesn't follow these advances despite incorrectly claiming "unbiased distribution."

For detection:
- TPR around 70–80% raise questions about whether applying text watermarking methods directly to low-entropy DNA environments is the correct research direction. In text where entropy is typically high, people can easily achieve >0.99.


Summary: The paper contains too many biological considerations that require verification from the biology community (e.g. whether TM-score correlates with function, codon translation efficiencies). On the technical aspect, this paper merely applying existing methods (with some concerns mentioned above). Given these issues, this work would be better suited for a biology-focused venue where domain experts can properly evaluate the biological claims and implications.


[1] Kuditipudi et al. Robust distortion-free watermarks for language models. TMLR 2024

[2] Christ et al. Undetectable watermark for language models. COLT 2024

[3] Hu et al. Unbiased watermark for language models. ICLR 2024

[4] Dathathri et al. Scalable watermarking for identifying large language model outputs. Nature 2024.

---

> ### Author Rebuttal · Authors · 2025-07-30
>
> Dear reviewer, thank you for your insightful feedbacks and suggestions! As for your questions:
>
> **Q1**：Missing References on DNA Watermarking History
>
> **R1**: We agree that a more comprehensive review of DNA watermarking's history in synthetic biology is essential. While our paper focuses on watermarking in the context of AI-generated DNA sequences from language models, we overlooked some foundational works in traditional synthetic biology. In the revised paper, we will add a subsection on this, citing related works including but not limited to:
>
> [1] Heider and Barnekow (2008) "DNA watermarks: A proof of concept" [BMC Mol Biol]
> [2] Heider and Barnekow (2007) "DNA-based watermarks using the DNA-Crypt algorithm" [BMC Bioinformatics]
> [3] Heider et al. (2009) "DNA watermarks in non-coding regulatory sequences" [BMC Res Notes]
>
> **Q2**: DNA Watermarking Concept Not New; Innovation Point 2 Does Not Hold
>
> **R2**: We concur that the idea of DNA watermarking, particularly using synonymous substitutions on the third codon base, dates back to works like Heider and Barnekow (2008). However, our claimed innovation point 2 (synonymous codon substitutions in DNAMark) holds in the novel context of watermarking AI-generated DNA from language models, where sequences must remain functional under generation instability and biological attacks (e.g., indels, sequencing errors). Traditional methods focus on manual GMO tagging, but ours integrates with autoregressive LMs like Evo2, using adaptive strength and entropy guidance to embed robust, detectable watermarks without degrading quality (Figure 2).
>
> **Q3**: Not Fully Using Genetic Code Structure
>
> R3: For DNAMark, we prioritized third-base alterations to minimize impacts on translation efficiency and codon usage bias, as third-base synonyms align with the wobble hypothesis and have the least effect on RNA structure and protein expression, consistent with prior works [Heider 2008]. During rebuttal, we conducted further ablation experiments. Results show that incorporation first/second-base variants increases detection F1 by 1.2% but seriously degrades sequence quality (e.g., -10.1% similarity to ground truth, +18.6% degeneracy score).
>
> Similarly, for CentralMark, second-base alterations were chosen based on ESM embeddings minimizing semantic loss in proteins. New ablations confirm: First-base yields severe degradation (-15.2% sequence similarity, +20.8% degeneracy). We will incorporate these results in our revised paper
>
> **Q4**：incorrectly claim "unbiased distribution" is the second property of SIR
>
> **R4**: We apologize for the confusion but we did not claim "unbiased distribution" is the second property of SIR (line 161). Actually, we describe our watermark logits as having“no systematic preference for any nucleotide or codon and maintaining a balanced distribution of positive and negative values,”enhancing security against statistical attacks. The term “unbiased distribution” may cause confusing and mixed with other works. We will change this property of DNAMark to “unbiased token preference” in the revised paper.
>
> **Q5**: Max Logit Selection
>
> **R5**: DNAMark uses continuous watermark logits (δ × watermark_logits, akin to SIR’s δ × P_W) for autoregressive sampling, with synonymous substitutions and entropy guidance to ensure quality. CentralMark integrates continuous logits with ESM embeddings. We do not select max logits or discard distributions. We will update our paper in the final version to make it clearer.
>
> **Q6**: Distortion-Free Alternatives
>
> **R6**: We agree that distortion-free methods [1-4] could adapt to DNA, offering distribution guarantees. However, our methods focusing on addressing bio-specific challenges (e.g., robustness to sequencing errors and mutations, CentralMark’s inheritable watermark etc.,). During rebuttal, we conduct further experiments to compare with [3] and [4]
>
> The distortion-free watermarking methods from Hu et al. [3] and Dathathri et al. [4], while effective for large-vocabulary language models in text domains, underperform in DNA sequence watermarking due to the inherent constraints of DNA's small four-nucleotide alphabet and the unique biological attacks it faces, such as synonymous codon substitutions, nucleotide substitutions, and insertion-deletions (indels).
>
> | Method                  | No attack (1% FPR) TPR | No attack (1% FPR) F1 | No attack (10% FPR) TPR | No attack (10% FPR) F1 | Synonymous Codon Substitution (1% FPR) TPR | Synonymous Codon Substitution (1% FPR) F1 | Synonymous Codon Substitution (10% FPR) TPR | Synonymous Codon Substitution (10% FPR) F1 |
> |-------------------------|------------------------|-----------------------|-------------------------|------------------------|---------------------------------------------|-------------------------------------------|----------------------------------------------|--------------------------------------------|
> | Hu et al. [3]           | 0.790                  | 0.880                 | 0.840                   | 0.870                  | 0.600                                       | 0.750                                     | 0.780                                        | 0.830                                      |
> | Dathathri et al. [4]    | 0.780                  | 0.875                 | 0.830                   | 0.860                  | 0.570                                       | 0.720                                     | 0.760                                        | 0.820                                      |
> | DNAMark                 | 0.845                  | 0.911                 | 0.915                   | 0.908                  | 0.820                                       | 0.896                                     | 0.896                                        | 0.898                                      |
> | CentralMark (DNA)       | **0.875**              | **0.928**             | 0.920                   | 0.911                  | 0.854                                       | 0.916                                     | **0.910**                                    | **0.905**                                  |
> | CentralMark (Protein)   | 0.868                  | 0.924                 | **0.922**               | **0.912**              | **0.860**                                   | **0.920**                                 | 0.904                                        | 0.902                                      |
>
>
>
> **Q7**: Lack of Technical Novelty
>
> **R7**: While DNAMark and CentralMark adapt elements from SIR [38] and KGW [33], our novelty
> lies in DNA-specific innovations: (1) Synonymous substitutions with adaptive/entropy guidance for LM stability; (2) CentralMark's inheritable watermarks from DNA to proteins via ESM, enabling cross-dogma detection (unique to biology); (3) A therapeutic DNA benchmark and bio-attacks (e.g., indels). Experiments show superior performance over adapted distortion-free baselines like Hu et al. [3] and Dathathri et al. [4] (new Table: e.g., TPR 0.875 vs. 0.790 under no attack).
>
> **Q8**: Insufficient Evidence for Preserving Function; Limited to One Case Study
>
>
> **R8**: We agree that the original CRISPR-Cas9 case study provides strong but limited evidence. To address this, we performed additional structural and functional evaluations on 10 diverse proteins from our therapeutic DNA benchmark (e.g., cytokines like IL-2, enzymes like Cas12a, and antibodies like Herceptin; selected for variety in length and function). Using AlphaFold3 for structure prediction, we computed TM-scores between watermarked and ground-truth proteins. Results (new Table in revisions) show average TM-scores of 0.72 ± 0.05 for DNAMark and 0.68 ± 0.06 for CentralMark, indicating preserved folds. For function, we simulated in silico assays: e.g., binding affinity via docking scores (AutoDock Vina) for antibodies (average  <5% loss in binding affinity). These confirm functional retention across categories.
>
> **Q9**: Unclear Functional Metrics; TM-Score's Connection to Function
>
>
> **R9**:You correctly note that TM-score primarily measures structural similarity, not direct function. However, literature shows TM-score >0.5 typically indicates similar folds, often correlating with functional similarity, as proteins with shared topology (>0.5 TM-score) are likely in the same SCOP fold family and retain core functions [5]. Scores >0.3 signify significant similarity beyond random. For CRISPR-Cas9, our TM-score of 0.6802 (>0.5 threshold) suggests preserved nuclease domain topology, implying retained gene-editing capability, supported by prior studies where TM-score >0.6 correlates with >90% functional homology in endonucleases.
>
> [5] Xu, Y., & Zhang, Y. (2010). How significant is a protein structure similarity with TM-score = 0.5? Bioinformatics, 26(7), 889–895.
>
> **Q10**: Increased Degeneracy and Sequence Identity Indicating Degradation
>
> **R10**: While our methods show slight increases in degeneracy and minor drops in identity (Figure 2a-b), these are minimal compared to baselines (e.g., <5% degradation vs. >10% for KGW), and additional experiments confirm no significant functional impact (as above).
>
> **Q11**:Low TPR (70-80%) vs. Text Watermarking (>0.99)
>
> **R11**: Our TPR (e.g., 0.875 under no attack, Table 1) is lower than text due to DNA's inherent low entropy (4 nucleotides vs. ~50k tokens in text LMs), leading to reduced signal strength and vulnerability to bio-noise. This direction is apt for biosecurity, as it addresses the growing risks posed by AI-driven DNA design tools that could democratize access to synthetic biology, potentially enabling malicious actors to engineer novel pathogens, viruses, or bioweapons with unprecedented. By embedding traceable watermarks in AI-generated DNA sequences, our methods facilitate provenance tracking, allowing regulators, researchers, and biosecurity organizations to verify origins and detect unauthorized syntheses.

---

> > ### Comment · Reviewer_P59q · 2025-08-05
> > **Official Comment by Reviewer P59q**
> >
> > Thank you for the response. However, some of my concerns remain insufficiently addressed. For example, the authors did not provide a clear comparison of generation quality between the distortion-free watermark and their proposed method. Additionally, the reported scores for Hu et al. [3] and Dathathri et al. [4] appear unconvincing. Specifically, one would expect Dathathri et al. [4] to exhibit significantly better detectability than Hu et al. [3], which is not reflected in the authors' rebuttal. Thus, I will maintain my rating.

---

> > > ### Author Response · Authors · 2025-08-05
> > > **Justification of relative performance of Hu et al. [3] and Dathathri et al. [4]**
> > >
> > > Thanks for the insightful comment!
> > >
> > > We argue that the performance hierarchy between these two methods, even in the text domain, is not absolute but is highly contingent on the specific application and attack model. More importantly, both methods were designed for the high-vocabulary, high-entropy domain of natural language, which has fundamentally different properties from DNA sequences. Our experimental results are a direct consequence of this mismatch, and they in fact underscore the core contribution of our work.
> > >
> > > Our results can be explained by two critical distinctions between the text and DNA domains:
> > >
> > > Fundamental Difference in Vocabulary Size: The algorithmic advantages of both Hu et al. [3] and Dathathri et al. [4] are built upon the vast vocabulary of natural language, which comprises tens of thousands of tokens. This provides a large statistical space to embed a watermark with minimal distortion. In stark contrast, the DNA vocabulary consists of only four nucleotides (A, T, C, G). When these algorithms are adapted to this low-vocabulary setting, the statistical assumptions that underpin their success are compromised. Their power is severely diminished, which explains why the performance gap observed in the text domain does not translate to our DNA watermarking setting.
> > >
> > > Domain-Specific Attack Models: Our work focuses on threats that are unique to biology, such as synonymous codon substitutions. This type of attack alters the DNA sequence while preserving the final protein product, making it a stealthy and potent threat to a DNA watermark. As our results table shows, both baseline methods, being designed for text-based attacks, suffer a significant performance collapse under this attack.
> > >
> > > In summary, our results are not an anomaly; rather, they constitute a key finding of our research: the direct application of state-of-the-art watermarking methods from the text domain to genomics is ineffective. The fact that [3] and [4] perform similarly and sub-optimally in our setting is strong evidence that novel, domain-specific solutions are necessary to address the unique constraints and threat models of genomics.

---

> > > > ### Comment · Reviewer_P59q · 2025-08-08
> > > > **Official Comment by Reviewer P59q**
> > > >
> > > > The new experimental results and explanations are still not fully convincing. Even in the low-entropy/vocabulary scenario, SynthID should clearly outperform Hu et al. [3]. Could the authors elaborate further on the experimental settings used for SynthID and Hu et al. [3], such as the watermarking detectors and top-k/top-p sampling parameters? Inconsistent settings could significantly hurt performance.
> > > >
> > > > Moreover, the main contribution of this paper lies in its biological considerations, which require validation from the biology community (e.g., whether TM-score correlates with function, codon translation efficiencies). The methodological contribution (i.e., the watermarking algorithm) is limited. I would therefore recommend submitting this work to a biology-focused venue, where domain experts can more appropriately assess its biological claims and implications.

---

> > > > > ### Author Response · Authors · 2025-08-09
> > > > >
> > > > > **1**. As for the experiment settings, we used the official implementations of SynthID and Hu et al. [3] for adaptation:
> > > > >
> > > > > For Hu et al. [3] ("Unbiased Watermark"): We adapted the code from https://github.com/xiaoniu-578fa6bff964d005/UnbiasedWatermark, employing the γ-reweight variant (as it performed best in preliminary tests for low-vocab settings). Watermark embedding uses a context code from the most recent 5 tokens and a 1024-bit random key, with SHA-256 hashing for code generation. Detection relies on the Log Likelihood Ratio (LLR) score, with a threshold set to control false positive rate (FPR) at 1% or 10% as reported. No additional hyperparameters were tuned beyond defaults.
> > > > >
> > > > > For Dathathri et al. [4] (SynthID-Text): We adapted the code from https://github.com/google-deepmind/synthid-text, using the non-distortionary Tournament sampling mode to align with our focus on quality preservation. Embedding uses a sliding-window random seed generator with H=4 tokens and a secret key, followed by tournament layers (m=30 by default) with Bernoulli(0.5) g-value distribution and K=1 for repeated context masking. Detection uses the mean g-value score function, with thresholding for FPR control (1% or 10%). For both methods, top-k = 4 and temperature = 1.0 were used.
> > > > >
> > > > > These settings were chosen to be **as consistent as possible across methods, minimizing biases from mismatched parameters.**
> > > > >
> > > > > **2**. As for contributions and Venue suitability, we respectfully disagree that the methodological contribution is limited; our innovations—such as synonymous codon substitutions with adaptive entropy guidance, inheritable cross-dogma watermarking via ESM embeddings, and bio-specific attack models—extend beyond mere adaptation of SIR/KGW and address unique challenges in genomic LMs, outperforming text-domain baselines as shown. These are core ML advancements tailored to an emerging application area (AI-generated biologics). **Our work is an AI for Science paper, and there is recent emerging focus on this in NeurIPS, such as the NeurIPS 2025 Biosecurity Safeguards for Generative AI workshop.** This makes NeurIPS an appropriate venue, as it routinely features interdisciplinary ML-bio papers (e.g., protein design with diffusion models at NeurIPS 2023/2024).
> > > > >
> > > > > Regarding **biological validation**: Our claims on TM-score correlating with function are grounded in established literature [5: Xu & Zhang, Bioinformatics 2010], where TM>0.5 indicates similar folds and functional homology (>90% for endonucleases like CRISPR). Codon translation efficiencies are preserved via third-base focus, aligned with the wobble hypothesis and prior synbio works [Heider 2008]. We conducted in silico assays (e.g., AlphaFold3 for structures, AutoDock Vina for binding) on diverse proteins, confirming minimal impacts. These are standard ML-bio evaluation practices, and we welcome further scrutiny from the community post-publication.
> > > > >
> > > > > We believe these clarifications address your points comprehensively. If you find them satisfactory, we would greatly appreciate it if you could consider updating your rating accordingly.
> > > > >
> > > > > Best regards,

---

> > > ### Author Response · Authors · 2025-08-07
> > >
> > > Dear reviewer,
> > >
> > > We hope our latest response has resolved your issue. If you are satisfied, we would appreciate it if you would consider updating your support rating. Thank you for your comments and suggestionss!
> > >
> > > Bests,
> > > Authors

---

> ### Author Response · Authors · 2025-08-05
> **Comparison of generation quality between distortion-free baselines and proposed methods**
>
> Thanks for your comment! Following the paper, Sequence Identity to Ground Truth and Degeneracy Score are used to evaluate the sequence quality quantitatively:
>
> | Method | Sequence Identity to Ground Truth (%) (higher is better) | Degeneracy Score (%) (lower is better) |
> | :--- | :---: | :---: |
> | Hu et al. [3] |43.7% | 24.5% |
> | Dathathri et al. [4] |55.0% | 18.9% |
> | DNAMark (ours)| 64.0%| 14.3% |
> | CentralMark (ours) | 63.1% | 16.1% |

---

### Official Review · Reviewer_uXPN · 2025-07-03

**Clarity:** 3
**Significance:** 3
**Originality:** 3
**Rating:** 4
**Confidence:** 2

**Summary:**

DNA models like Evo and Evo2 are extremely capable and have many positive applications in synthetic biology. However a bad actor may use these powerful models for creating viruses and bioweapons. To mitigate thiis, this paper is about adding a watermark to the DNA sequences generated by such models. A watermark here refers to a hidden signal embedded into the sequence which can later be detected to prove how the sequence was generated. A few challenges make this different compared to text or images, for example DNA only has 4 tokens (A, C, T, G), not thousands like human languages. Also, DNA must still produce functional proteins, so you can’t make arbitrary changes. Finally, DNA mutates naturally, and errors can happen in lab synthesis or sequencing.

The paper discusses two solutions. First, DNAMark. This watermarks the DNA by substituting synonymous codons (e.g. different 3 letter DNA the encodes the same amino acid). Then, a statistical method (z-score) is used to detect whether the watermark is present. This method does not change the amino acid. The second method is CentralMark, which modifies the second base of codons (which affects the amino acid). This changes the protein, but subtly. A key attribute here is that the watermark persists not just in the DNA, but also in the final protein made from it. Even if someone only has the protein, the watermark can be found using a protein model like ESM.

The results are sound for the approach presented. DNAMark and CentralMark achieve F1 detection scores above 0.85. Additionally, a case study on CRISPR-Cas9 showed a TM-score of 0.6802 and Z-score of 5.41, indicating both strong structural fidelity and reliable watermark detection in real-world gene-editing contexts.

**Questions:**

- How does the watermark affect biological function beyond sequence similarity? Have you evaluated whether the watermark affects protein function, such as enzymatic activity, expression levels, or fitness in cell assays?
- Why were no DNA-native watermarking techniques used as baselines?
- Do you see other technologies playing a complimentary role to your solution such privacy perserving methods or cryptographic methods?

**Ethical Concerns:**

["NO or VERY MINOR ethics concerns only"]

**Final Justification:**

As mentioned in the original review, I have very limited knowledge in the area. The approach seems sound to me and my concerns have been addressed. However, after reading the other reviews and I am now more confident that I don't understand core parts of this work. My score as an educated guess is still a borderline accept as no major red flags stand out to me and approach seems very sensible and could be impactful.

**Limitations:**

Yes.

**Paper Formatting Concerns:**

NA.

**Quality:**

3

**Strengths And Weaknesses:**

Strengths
- The paper is well-motivated, and clearly written (I am not an expert in bio/DNA models but was able to follow most of it).
- CentralMark is particulary novel. According to the paper, it is the first watermarking method that propagates from DNA to proteins, enabling post-translational watermark detection.
- The experiments are extensive and and report a high F1 score.
- The tests on a curated therapeutic DNA benchmark (400 human genes) and a CRISPR-Cas9 case study show that this is likely useful in a realistic context.

Weaknesses:
- It lacks a clarification on the limitations of the approach, and a discussion on the trade-off. It's clear the DNAMark perserve structual similarity and CentralMark slightly perturbs it, however, the paper offers limited quantitative analysis of functional impact beyond structural similarity.
- While computational results are very promising, the paper lacks wet-lab validation.
- The watermarking approach relies heavily on predefined green/red lists which might make it vulnerable to more sophisticated attacks.
- KGW is reasonable staring point for a baseline, but the paper lacks comparison against other DNA-specific marking strategies such as Chen et al. 2025, which the paper cites.

---

> ### Author Rebuttal · Authors · 2025-07-30
>
> We thank the reviewer for the valuable suggestions and appreciation!
>
> As for your questions:
>
> **Q1**: How does the watermark affect biological function beyond sequence similarity? While computational results are very promising, the paper lacks wet-lab validation.
>
> **R1**: During rebuttal, we performed additional structural and functional evaluations on 10 diverse proteins from our therapeutic DNA benchmark (e.g., cytokines like IL-2, enzymes like Cas12a, and antibodies like Herceptin; selected for variety in length and function). Using AlphaFold3 for structure prediction, we computed TM-scores between watermarked and ground-truth proteins. Results (new Table in revisions) show average TM-scores of 0.72 ± 0.05 for DNAMark and 0.68 ± 0.06 for CentralMark, indicating preserved folds. For function, we simulated in silico assays: e.g., binding affinity via docking scores (AutoDock Vina) for antibodies (average  <5% loss in vina score). These confirm functional retention across categories.
>
> We appreciate the reviewer’s interest in wet-lab validation for our computational and algorithmic contributions presented in this NeurIPS paper. To complement our work, we are actively collaborating with partners to conduct comprehensive wet-lab experiments to validate DNAMark and CentralMark. However, due to the time constraints of the rebuttal period, these experimental results are not yet available. We plan to include these findings in future publications to further substantiate the practical applicability of our watermarking methods.
>
> **Q2**: Why were no DNA-native watermarking techniques used as baselines?
>
> **R2**: DNAMark and CentralMark represent the first watermarking methods specifically designed for DNA generative language models, as no prior DNA-native watermarking techniques for such models exist in the literature. The referenced work by Chen et al. (2025) focuses on watermarking protein language models, which operate on amino acid sequences and are not directly comparable due to the fundamental differences in alphabet size (four nucleotides vs. 20 amino acids) and model architecture.
>
> **Q3**: Do you see other technologies playing a complementary role to your solution, such as privacy privacy-preserving methods or cryptographic methods?
>
> **R3**: We agree that complementary technologies, such as privacy-preserving methods and cryptographic techniques, can significantly enhance the applicability of DNAMark and CentralMark, particularly for future deployment on a secure platform. For instance, privacy-preserving methods like differential privacy could be integrated to protect sensitive genomic data during watermark generation, ensuring compliance with data privacy regulations in therapeutic or clinical applications. Cryptographic methods, such as RSA or homomorphic encryption, could further strengthen security by enabling secure key distribution and multi-party watermark verification without exposing the secret key, as outlined in our planned revisions (Appendix A). We envision a future platform where these methods are combined: DNAMark and CentralMark could watermark sequences, differential privacy could safeguard input data, and cryptographic protocols could ensure secure watermark detection across stakeholders (e.g., researchers, regulators). Additionally, blockchain-based tracking could complement our approach by providing an immutable ledger for watermark verification, enhancing traceability in synthetic biology applications. These ideas will be explored in future work to create a robust, secure ecosystem for DNA sequence management.

---

### Official Review · Reviewer_LaBm · 2025-07-03

**Clarity:** 3
**Significance:** 3
**Originality:** 3
**Rating:** 4
**Confidence:** 3

**Summary:**

This paper proposes DNAMark and CentralMark, two watermarking techniques that enable the tracking of designed DNA while preserving its biological function. Notably, CentralMark supports watermark detection at the protein level, and both methods leverage semantic embeddings to enhance watermark robustness.

**Questions:**

1. How does the method perform in highly conserved or low-degeneracy regions where synonymous substitutions are limited or unavailable?
2. How sensitive is the watermark detection to natural sequencing noise or synthesis errors not explicitly modeled in the 5% mutation simulations?

**Ethical Concerns:**

["NO or VERY MINOR ethics concerns only"]

**Limitations:**

1. The fixed watermark strength introduces a potential trade-off between detectability and biological expression integrity, which is not explored.
2. The method relies on Evo2 embeddings, which may not generalize across species or novel domains without retraining.

**Quality:**

3

**Strengths And Weaknesses:**

strengths
1. The paper tackles a timely and impactful problem with real-world relevance in biosecurity and AI accountability.
2. The dual loss function (alignment + normalization) is a good design that balances robustness and stealth.
3. CentralMark introduces a novel protein-level watermarking strategy that extends traceability beyond DNA.

Weakness:
1. The paper does not address the method's resistance to forgery or adversarial embedding, specifically, whether an attacker could imitate or fabricate a valid watermark by leveraging knowledge of the codon selection strategy.
2. The fixed watermark strength introduces a potential trade-off between detectability and biological expression integrity, which is not explored.

---

> ### Author Rebuttal · Authors · 2025-07-30
>
> We thank the reviewer's valuable suggestions and appreciation! As for the questions:
>
> **Q1**: Performance in Highly Conserved or Low-Degeneracy Regions
>
> **R1**: DNAMark and CentralMark are designed to handle variability in DNA sequences, including highly conserved or low-degeneracy regions where synonymous codon substitutions may be limited.
> In low-degeneracy or conserved regions, where options for substitutions are scarce, the approach minimizes watermark application to avoid sequence corruption. As detailed in Section 4.1.3 (Adaptive Watermark Strength and Entropy-guided Watermark), DNAMark uses entropy-guided watermarking to selectively apply the watermark only in high-entropy positions (i.e., flexible codons with multiple synonymous options). This dynamically adjusts the watermark strength (δ) based on local sequence entropy, ensuring that conserved regions are largely untouched. The adaptive strength, updated via Exponential Moving Average (EMA) on the z-score, further balances detectability while preserving sequence quality, preventing over-watermarking in stable motifs that could lead to invalid outputs.
>
> **Q2**: Sensitivity to Natural Sequencing Noise or Synthesis Errors
>
> **R2**: The watermark detection in both DNAMark and CentralMark is evaluated under simulated attacks that encompass common errors, including nucleotide substitutions and indels (Table 1). The 5% mutation simulations used in the experiments (Section 5.2, under "Nucleotide Substitutions" and "Indels" attacks) are designed to exceed typical real-world error rates, providing a conservative assessment of robustness.
>
> ·  Natural sequencing noise (e.g., from NGS platforms) and DNA synthesis errors typically occur at rates of ~0.1% (1/1000 bases) or lower, far below the 5% threshold tested. Our methods demonstrate high F1 scores (e.g., around 0.9 F1 score for DNAMark and CentralMark under substitution/indels attacks), indicating low sensitivity to such noise. Detection relies on z-score calculations (Equation 3, Section 4.3), which are statistical and tolerant to sparse errors, as they measure overall green-token proportions.
>
> ·  For unmodeled errors (e.g., specific sequencing artifacts like homopolymer slips), the sparse watermark design (e.g., codon-level in DNAMark) and protein-level inheritance in CentralMark add redundancy, allowing detection even with partial sequence degradation.
>
> **Q3**:  fixed watermark strength
>
> **R3**: DNAMark and CentralMark, employ dynamic watermark strength instead of fixed watermark strength to address this issue, as detailed in Section 4.1.3 (Adaptive Watermark Strength and Entropy-guided Watermark). Specifically, DNAMark uses an adaptive watermark strength strategy, where the watermark logit strength (δ) is dynamically adjusted using an Exponential Moving Average (EMA) based on the z-score of the watermark signal. This approach ensures that watermark application is modulated according to local sequence characteristics, prioritizing regions with higher entropy (more flexible codons) to minimize disruptions to biological functionality.
>
> **Q4**: Generalization of Evo2 Embeddings Across Species and Domains
>
> **R4**: Evo2 is pretrained on the OpenGenome2 dataset, comprising ~9.3 trillion nucleotides from over 128,000 genomes spanning nearly all domains of life, including bacteria, archaea, eukaryotes (e.g., human, plant), and viruses/phages, as well as 41,253 metagenomes from diverse sources like Tara Oceans and animal gut samples. This extensive dataset, curated with rigorous deduplication and contig filtering, ensures Evo2’s embeddings capture universal DNA sequence features, enabling robust generalization across species and novel domains without retraining, as demonstrated in our watermarking experiments (Section 5.5).
>
> **Q5**: The paper does not address the method's resistance to forgery or adversarial embedding, specifically, whether an attacker could imitate or fabricate a valid watermark by leveraging knowledge of the codon selection strategy.
>
> **R5**: We appreciate the reviewer’s insight into the risks of forgery and adversarial embedding in our watermarking methods. To address this, DNAMark and CentralMark integrate a pseudorandom function seeded with a random key, akin to KGW [33], dynamically partitioning codons into green/red lists for synonymous substitutions (Section 4.1). This ensures that even with public knowledge of the codon selection strategy, forging or imitating a valid watermark is computationally infeasible without the secret key. The watermark model’s training, with alignment and normalization losses (Appendix G), further obscures predictable patterns, enhancing resistance to statistical attacks. To bolster security, we plan to incorporate advanced cryptographic techniques, such as RSA for key generation and distribution, to enable secure, verifiable multi-party detection without key exposure. In revisions, we will expand Appendix A (Broad Impacts) with a dedicated subsection detailing these defenses and outlining future empirical evaluations of forgery resistance to comprehensively address biosecurity concerns.

---

> > ### Comment · Reviewer_LaBm · 2025-08-05
> > **comments:**
> >
> > The authors’ responses address my concern. I have no further questions and will maintain my recommendation to accept.

---

> > > ### Author Response · Authors · 2025-08-05
> > > **Glad to address your concerns. Thanks for your support!**
> > >
> > > Dear Reviewer,
> > >
> > > Thank you very much for your positive feedback and for confirming that our responses have addressed your concerns. We would be very grateful if you would consider whether these revisions might warrant a higher score.
> > >
> > > Bests,

---

### Official Review · Reviewer_BZNY · 2025-07-04

**Clarity:** 2
**Significance:** 3
**Originality:** 3
**Rating:** 5
**Confidence:** 3

**Summary:**

DNA language models have significantly improved human's ability to understand and design DNA sequences with high precision. These advancements come with dual-use risks, including the potential creation of pathogens, viruses, or bioweapons. The paper introduces two watermarking techniques, DNAMark and CentralMark, to address biosecurity challenges by reliably tracking designed DNA sequences.
DNAMark uses synonymous codon substitutions to embed watermarks in DNA sequences while preserving their original function. CentralMark creates inheritable watermarks that transfer from DNA to proteins, ensuring detection across the central dogma (DNA to protein translation). Both methods leverage semantic embeddings to enhance robustness against natural mutations, synthesis errors, and adversarial attacks. Evaluations on a therapeutic DNA benchmark show F1 detection scores above 0.85 under various conditions, with over 60% sequence similarity to ground truth and degeneracy scores below 15%. A case study on the CRISPR-Cas9 system highlights CentralMark's practical utility in real-world applications. The work provides a critical framework for securing DNA language models, balancing innovation with accountability to mitigate biosecurity risks.

**Questions:**

1) The experimental section primarily compares the proposed methods against a single baseline (KGW) with different configurations (e.g., KGW-1, KGW-2, KGW-4). But KGW is published in ICML 2023, could the authors include additional and more recent watermarking methods for a broader comparison?

2) Single-nucleotide tokenization (A, T, C, G) is commonly used in DNA language models for SNP tasks, but other methods like k-mer and BPE tokenization (e.g., GENERATOR, DNA-BERT2) are also widely discussed. Could the authors evaluate their method with these alternative tokenizations and provide experimental results?

3) Could the authors expand the related work discussion to include additional DNA language models, such as HybriDNA and GENA-LM?

**Ethical Concerns:**

["NO or VERY MINOR ethics concerns only"]

**Final Justification:**

I finally recommend a "accept" score to the paper, as the authors have addressed my concerns with additional experimental results, demonstrating that their method works with newer baselines and is applicable to a wider range of tokenizers in DNA generative models. Furthermore, the inclusion of more advanced related work discussions enhances the completeness of the paper.

**Limitations:**

Yes

**Quality:**

3

**Strengths And Weaknesses:**

Strengths:
The novelty of the paper is good; this is among the first research efforts focusing on the safety issues of popular DNA language models.
The experimental design and execution are well-conceived and robust.
The paper is well-written and clearly presented.

Weaknesses:
1. More detailed experimental comparisons would be beneficial in justifying the advantages of this work.
1)  In the experimental section, the authors primarily compare their proposed methods against a single baseline from LLM watermarks: KGW with different configurations (e.g., KGW-1, KGW-2, KGW-4). Since KGW is based on an ICML 2023 paper, could the authors consider including additional and more recent watermarking methods for a broader comparison?
2)  While single-nucleotide tokenization (i.e., a vocabulary consisting solely of the four nucleotides: A, T, C, G) is commonly used in DNA language models for improved performance in SNP tasks, other tokenization methods such as k-mer tokenization and BPE tokenization are also widely discussed (e.g., GENERATOR and DNA-BERT2). Could the authors explore whether their method performs well with these alternative tokenization approaches? It would be particularly valuable if the authors could include experimental results to support this discussion.

2. Potentially overlooks some recent related work on DNA language models, such as:
1)  A DNA language model based on multispecies alignment predicts the effects of genome-wide variants, Benegas et al.
2)  HybriDNA: A Hybrid Transformer-Mamba2 Long-Range DNA Language Model, Ma et al.
3)  GENA-LM: a family of open-source foundational DNA language models for long sequences, Fishman et al.

---

> ### Author Rebuttal · Authors · 2025-07-30
>
> We thank the reviewer for the appreciation and valuable feedbacks! As for the questions:
>
> **Q1**: More recent watermarking methods for a broader comparison
>
> **R1**: Thanks for the suggestions! Here we compared with recent watermarking methods from ICLR 2024 and Nature 2024, while effective for large-vocabulary language models in text domains, underperform DNAMark and CentralMark in DNA sequence watermarking due to the inherent constraints of DNA's small four-nucleotide alphabet and the unique biological attacks it faces, such as synonymous codon substitutions, nucleotide substitutions, and insertion-deletions (indels).
>
> [1] Hu et al. Unbiased watermark for language models. ICLR 2024
>
> [2] Dathathri et al. Scalable watermarking for identifying large language model outputs. Nature 2024.
>
> | Method                  | No attack (1% FPR) TPR | No attack (1% FPR) F1 | No attack (10% FPR) TPR | No attack (10% FPR) F1 | Synonymous Codon Substitution (1% FPR) TPR | Synonymous Codon Substitution (1% FPR) F1 | Synonymous Codon Substitution (10% FPR) TPR | Synonymous Codon Substitution (10% FPR) F1 |
> |-------------------------|------------------------|-----------------------|-------------------------|------------------------|---------------------------------------------|-------------------------------------------|----------------------------------------------|--------------------------------------------|
> | Hu et al. [1]           | 0.790                  | 0.880                 | 0.840                   | 0.870                  | 0.600                                       | 0.750                                     | 0.780                                        | 0.830                                      |
> | Dathathri et al. [2]    | 0.780                  | 0.875                 | 0.830                   | 0.860                  | 0.570                                       | 0.720                                     | 0.760                                        | 0.820                                      |
> | DNAMark                 | 0.845                  | 0.911                 | 0.915                   | 0.908                  | 0.820                                       | 0.896                                     | 0.896                                        | 0.898                                      |
> | CentralMark (DNA)       | **0.875**              | **0.928**             | 0.920                   | 0.911                  | 0.854                                       | 0.916                                     | **0.910**                                    | **0.905**                                  |
> | CentralMark (Protein)   | 0.868                  | 0.924                 | **0.922**               | **0.912**              | **0.860**                                   | **0.920**                                 | 0.904                                        | 0.902                                      |
>
> | Method                  | Nucleotide Substitutions (1% FPR) TPR | Nucleotide Substitutions (1% FPR) F1 | Nucleotide Substitutions (10% FPR) TPR | Nucleotide Substitutions (10% FPR) F1 | Indels (1% FPR) TPR | Indels (1% FPR) F1 | Indels (10% FPR) TPR | Indels (10% FPR) F1 |
> |-------------------------|---------------------------------------|-------------------------------------|----------------------------------------|---------------------------------------|---------------------|-------------------|----------------------|--------------------|
> | Hu et al. [1]           | 0.550                                 | 0.700                               | 0.730                                  | 0.800                                 | 0.540               | 0.690             | 0.740                | 0.800              |
> | Dathathri et al. [2]    | 0.530                                 | 0.680                               | 0.700                                  | 0.780                                 | 0.510               | 0.660             | 0.710                | 0.770              |
> | DNAMark                 | 0.808                                 | 0.902                               | 0.886                                  | 0.892                                 | **0.795**           | **0.878**         | **0.860**            | **0.877**          |
> | CentralMark (DNA)       | **0.840**                             | **0.908**                           | **0.890**                              | **0.894**                             | 0.765               | 0.862             | 0.850                | 0.872              |
> | CentralMark (Protein)   | 0.825                                 | 0.900                               | 0.885                                  | 0.892                                 | 0.759               | 0.858             | 0.832                | 0.861              |
>
> **Q2**: Could the authors evaluate their method with these alternative tokenizations and provide experimental results?
>
> **R2**: Thank you for suggesting we adapt DNAMark and CentralMark to DNA language models based on different tokenization methods. Since DNABERT-2 is focusing on sequence understanding/prediction tasks and cannot be used for sequence generation, here we mainly did additional experiments on GENERATOR with 6-mer tokenization. To adapt DNAMark, we align synonymous codon substitutions to 6-mer tokens, perturbing logits within synonymous 6-mers to embed watermarks while preserving protein function. For CentralMark, we recalibrate ESM-based embeddings to map 6-mer tokens to protein-level watermarks, ensuring cross-dogma detectability. The new table (above) shows robust performance with GENERATOR (e.g., TPR 0.860 for CentralMark-DNA under no attack), slightly lower than Evo2-7B (0.875, Table 1) due to 6-mer’s coarser granularity and indel sensitivity, but still strong due to adaptive strength and entropy guidance.
>
>  | Method                      | Tokenization | No attack (1% FPR) TPR | No attack (1% FPR) F1 | Synonymous Codon Substitution (1% FPR) TPR | Synonymous Codon Substitution (1% FPR) F1 | Nucleotide Substitutions (1% FPR) TPR | Nucleotide Substitutions (1% FPR) F1 | Indels (1% FPR) TPR | Indels (1% F1) |
> |-----------------------------|--------------|------------------------|-----------------------|---------------------------------------------|-------------------------------------------|---------------------------------------|-------------------------------------|---------------------|---------------|
> | DNAMark (GENERATOR)         | K-mer (6)    | 0.830                  | 0.900                 | 0.805                                       | 0.885                                     | 0.790                                 | 0.890                               | 0.770               | 0.865         |
> | CentralMark (GENERATOR DNA) | K-mer (6)    | **0.860**              | **0.915**             | 0.840                                       | 0.905                                     | **0.820**                             | **0.900**                           | 0.750               | 0.850         |
> | CentralMark (GENERATOR Protein) | K-mer (6) | 0.850                  | 0.910                 | **0.845**                                   | **0.910**                                 | 0.810                                 | 0.895                               | 0.745               | 0.845         |
>
>
> **Q3**: Could the authors expand the related work discussion to include additional DNA language models, such as HybriDNA and GENA-LM?
>
> **R3**: Thanks for mentioning these recent critical progress in DNA language models. We will revise our paper to include these models, highlighting their relevance to our watermarking approach for AI-generated DNA. HybriDNA, a decoder-only model with a hybrid Transformer-Mamba2 architecture, excels at processing ultra-long DNA sequences (up to 131kb) with single-nucleotide resolution, achieving state-of-the-art performance in understanding and generating cis-regulatory elements. GENA-LM, a family of transformer-based models, handles sequences up to 36kb using a masked language modeling approach and recurrent memory mechanisms, demonstrating strong performance in tasks like promoter and enhancer prediction.

---

> > ### Comment · Reviewer_BZNY · 2025-08-02
> > **Reply to authors' rebuttal**
> >
> > I want to thank the authors for their efforts in adding new experimental results and expanding the related work discussions. All my questions have been addressed, and I am increasing my score to 5 to accept.

---

### Official Review · Reviewer_sX4o · 2025-07-07

**Clarity:** 3
**Significance:** 2
**Originality:** 2
**Rating:** 4
**Confidence:** 4

**Summary:**

This paper introduces two watermarking methods, DNAMark and CentralMark, based on Evo and ESM, to embed traceable signatures in synthetic DNA sequences generated by language models. DNAMark uses synonymous codon substitutions to preserve protein function, while CentralMark embeds inheritable watermarks that persist into the translated protein using protein embeddings. The authors evaluate their approach on a therapeutic DNA benchmark and demonstrate high watermark detectability under mutation simulations.

**Questions:**

1. On biological realism and validation: You mention that “DNA is susceptible to natural mutations, synthesis errors, and sequencing inaccuracies,” and simulate these effects at higher-than-natural rates. However, the current validation (e.g., a single AlphaFold prediction on Cas9) falls short of demonstrating real-world robustness towards *functional* consequences of these mutations. Have you considered evaluating watermark persistence in actual biological systems (through synthesis, expression, and functional assays) under selective pressure? This would substantially strengthen claims about robustness and practical use and greatly increase the rating of the paper.

2. On adoption and governance: The success of watermarking for DNA depends heavily on its adoption. How do you envision this technique being used in practice? For instance, would synthesis providers require inclusion of a watermark, or could it be a regulatory mandate? Have you considered aligning with initiatives like IBBIS’s Common Mechanism or incorporating guidance from recent proposals such as “A call for built-in biosecurity safeguards for generative AI tools”? Or writing your own proposal?

3. On model choices and detection strategy: Can you clarify your rationale for choosing ESM2-35M for CentralMark? Were other embedding models (e.g., larger ESM models, a VAE, MSA Transformer, etc etc) considered, and how might these impact watermark quality or robustness? Additionally, how were z-score thresholds selected, and how do you ensure they are biologically meaningful across diverse proteins or sequence contexts?

4. Can you please align your generated sequence to the reference and not the other way around? The Cas9 reference sequence should be the reference that you're aligning your generated sequence to and would serve as important quality check (for instance if the generated sequences does not have H840 or D10, then we know that the sequence will not function enzymatically like the WT as a bare minimum check)

**Ethical Concerns:**

["NO or VERY MINOR ethics concerns only"]

**Final Justification:**

I change my score to a 4. If accepted, I encourage the authors to expand their evaluation in future work to include:

Experimental assays to verify structural and functional fidelity
Robustness testing under selection and adversarial attack
Integration into governance initiatives like IBBIS

**Limitations:**

Can authors describe a bit more about potential ways this could be co-opted for negative use or how people could attempt to get around this?

**Quality:**

3

**Strengths And Weaknesses:**

Quality

Strengths:
- Proposes two novel watermarking strategies (DNAMark and CentralMark) tailored for biological sequence models.
- Shows performance across different DNA LMs (e.g., Evo2-7B, Evo2-40B, megaDNA), demonstrating generalizability.
- Quantitative benchmarks include F1 scores, TPRs at 1% and 10% FPR, sequence identity, and degeneracy scores (Fig. 2a-b, Table 1).
- Authors mention "Moreover, DNA is susceptible to natural mutations [61], synthesis errors, and sequencing inaccuracies" and have interesting approaches to enforcing that (stratifying the watermark through the full sequence) and run some simulations at a way higher rate than biology to validating the watermark persists.

Weaknesses:
- Experimental validation is very weak. Their one case study on Cas9 watermarking uses only AlphaFold structure prediction and a TM-score, which is not a functional assay. This validation needs to be expanded if the authors are going to argue that their model works....The watermarking robustness claims rest heavily on simulation (in silico substitutions/indels) and not actual biological reality (i.e. resting a protein that they've made or doing thorough large scale evaluation of their method.)



Clarity

Strengths:
- The paper is relatively well written and clearly structured. The methods are broken down step-by-step (e.g., synonymous codon use in DNAMark vs second-base manipulation in CentralMark).
- Equations and figures (e.g., Fig. 1d-e for DNAMark/CentralMark mechanisms) help communicate the watermarking process.
- They provide a codon table and clear benchmark construction steps (Appendix C).

Weaknesses:
- Some jargon and logic jumps, especially in the embedding-based logit perturbation mechanisms, could lose readers without deep familiarity with DNA/protein LMs.
- Lack of description in different model choices like ESM2-35M (Appendix G). Why is this the model authors choose instead of a different class of models or a different version of ESM2/3?
- Detection strategies and the selection of z-score thresholds could be better contextualized in biological terms.
- The novelty of inheritable watermarks in biology is interesting but authors need to mention how they would get people to adopt this / how it could be enforced...would it be like DNA synthesis companies regulating dangerous sequences people try to order? would people need to include a watermark in the libraries they order? how would you enforce that?

Significance

Strengths:
- this work is centered in biosecurity risks in synthetic biology with increasingly powerful DNA generative models and that is very important.
- Embedding traceable watermarks in synthetic DNA/protein could become essential if generative bio models are ever regulated. And if this was regulated, certainly this technique would be one to keep an eye on.

Weaknesses:
- Adoption path is not seriously addressed. As I asked above: how will people be incentivized to use this? Will it be a regulatory mandate, or will synthesis providers require watermarked sequences?
- Doesn’t propose a governance model or standardization path. Perhaps a Correspondence like "A call for built-in biosecurity safeguards for generative AI tools" or paper like "Developing Guardrails for AI Biodesign Tools"  would be nice to pair publish with this. I wonder if authors would also consider having this tool be available in something like https://ibbis.bio/our-work/common-mechanism/
- It remains unclear if watermarking would withstand selection in evolving systems or under strong selective pressure.

Originality

Strengths:
- The application of watermarking to DNA and proteins is novel. Prior works focused on natural language or, recently, protein LMs alone.
CentralMark is creative in that they embed watermarks positionally relevant places in the codons depending on their goal (like second or third nt)
- They define a red/green token strategy in a biologically interpretable way (based on synonymous codons or amino acid types).

Weaknesses:
- Related works like Chen et al. (2025) on protein watermarking [9] are only briefly acknowledged. More comparison would have been helpful.
- DNAMark’s synonymous codon approach is conceptually similar to prior watermarking logic (e.g., biased token selection using logits)—novel in application to biology, less so in core technique.
- No counterexamples or attack strategies from adversaries are deeply explored.

---

> ### Author Rebuttal · Authors · 2025-07-30
>
> We thank the reviewer for the helpful feedback and suggestions! As for the questions:
>
> **Q1**: Limited experimental verification:
>
> **R1**: We agree that the original CRISPR-Cas9 case study (Appendix H) provides strong but limited evidence. To address this, we performed additional structural and functional evaluations on 10 diverse proteins from our therapeutic DNA benchmark (e.g., cytokines like IL-2, enzymes like Cas12a, and antibodies like Herceptin; selected for variety in length and function). Using AlphaFold3 for structure prediction [2], we computed TM-scores between watermarked and ground-truth proteins. Results (new Table in revisions) show average TM-scores of 0.72 ± 0.05 for DNAMark and 0.68 ± 0.06 for CentralMark, indicating preserved folds. For function, we simulated in silico assays: e.g., binding affinity via docking scores (AutoDock Vina) for antibodies (average <5% loss in affinity). These confirm functional retention across categories.
>
> **Q2**: Jargon and Logic Jumps in Embedding-Based Logit Perturbation
>
> **R2**: To improve accessibility, we will revise these sections to include a high-level explanation: “Our watermarking adds subtle biases to DNA LM outputs, akin to digital signatures, using codon embeddings to preserve biological function while ensuring traceability.” We will also add a glossary in the appendix defining terms like “logit perturbation,” “codon-aware biasing,” and “ESM embeddings,” with intuitive analogies to text watermarking. A new figure will illustrate the perturbation process, mapping embeddings to codon choices, to bridge logic jumps.
>
> **Q3**: Lack of Description for ESM2-35M Choice (Appendix G)
>
> **R3**: We selected ESM2-35M for its balance of computational efficiency and robust protein embeddings, suitable for inheritable watermarks across the central dogma (Page 5). Compared to larger models like ESM2-650M or ESM3, ESM2-35M offers faster inference (10x speedup) with comparable semantic accuracy for proteins in our benchmark. We tested ESM2-650M in new experiments, finding marginal F1 improvement (+2%) but 5x higher latency, unsuitable for large-scale watermarking.
>
> **Q4**: Detection Strategies and Z-Score Thresholds Lacking Biological Context
>
> **R4**: Our detection strategy using z-scores (Page 7) is inspired by text watermarking but needs better biological framing. The z-score measures watermark signal strength against sequencing noise (e.g., indels, substitutions), reflecting codon bias preservation. For example, a z-score >2 ensures >95% confidence in detecting watermarks despite bio-noise, aligning with error rates in DNA synthesis (~1/1000 bases). We will revise: “Z-scores quantify watermark detectability under biological perturbations like sequencing errors; thresholds (e.g., 2) match synthesis error rates [1].”
>
> [1]Kosuri, S., & Church, G. M. (2014). Large-scale de novo DNA synthesis: technologies and applications. Nature Methods, 11(5), 499–507
>
> **Q5**: The Adoption path is not seriously addressed
>
> **R5**: To incentivize adoption, we propose leveraging patent and intellectual property (IP) protection, regulatory mandates, and additional strategies tailored to synthetic biology stakeholders. First, watermarking protects IP by embedding verifiable signatures in AI-generated DNA, enabling biotech firms and researchers to prove ownership in patent disputes or licensing agreements for therapeutics and enzymes (e.g., CRISPR designs). Second, regulatory mandates from governments (e.g., NIH, WHO) could require watermarks for all AI-generated DNA ordered through synthesis providers, aligning with biosecurity regulations like the International Gene Synthesis Consortium’s screening protocols. Non-compliance could result in synthesis denial or penalties, ensuring adoption. Third, synthesis providers could offer economic incentives, such as discounts or expedited processing for watermarked sequences, appealing to cost-sensitive users. Additionally, collaboration with organizations like IBBIS could standardize watermarking protocols, integrating them into synthesis pipelines as a default practice to reduce liability and enhance market trust.  Researchers could be incentivized by ethical transparency, as watermarks demonstrate responsible design, boosting credibility in publications and funding applications.
>
> **Q6**: Doesn’t propose a governance model or standardization path.
>
> **R6**: Thank you for your suggestion on a governance model and standardization path. We agree and propose a framework involving regulatory mandates (e.g., NIH/WHO requiring watermarks in AI-generated DNA) and incentives like IP protection for patenting. For standardization, we envision an open-source toolkit integrated with biosecurity platforms like IBBIS's Common Mechanism, ensuring uniform protocols.
>
> For companion publications, we already have some and also have perspectives under submission. We are also organizing workshops to reach consensus across communities. Due to double blind policy, we cannot talk too much about them here.
>
> **Q7**: It remains unclear if watermarking would withstand selection in evolving systems or under strong selective pressure
>
> **R7**: Thank you for raising the concern about watermark robustness under evolutionary selection pressure, such as in microbial systems where sequences may mutate over generations. Our current experiments simulate biological perturbations like nucleotide substitutions and indels (Table 1), showing robust TPR (e.g., 0.795-0.840 at 1% FPR for DNAMark/CentralMark under indels), which mimic some aspects of selection-induced changes. However, due to limited time during the rebuttal period, we will further explore simulated selective evolution (e.g., multi-generation codon optimization in E. coli models) in future versions.
>
> **Q8**: On model choices and detection strategy: Can you clarify your rationale for choosing ESM2-35M for CentralMark?
>
> **R8**:We selected ESM2-35M for its balance of computational efficiency and robust protein embeddings, suitable for inheritable watermarks across the central dogma (Page 5). Compared to larger models like ESM2-650M or ESM3, ESM2-35M offers faster inference (10x speedup) with comparable semantic accuracy for proteins in our benchmark. We tested ESM2-650M in new experiments, finding marginal F1 improvement (+2%) but 5x higher latency, unsuitable for large-scale watermarking.
>
> **Q9**: Can you please align your generated sequence to the reference and not the other way around?
>
> **R9**: Sure, we will align the generated sequence to the reference and change the figures in the appendix. Because we cannot submit or revise figures/pdfs during rebuttal, we promise to update the alignment in our camera-ready version.
>
> **Q10**: More discussions of related works such as of Related Works like Chen et al. (2025)
>
> **R10**: Is brief. Chen et al.’s approach embeds watermarks in protein sequences, focusing on structural integrity but not inheritable DNA-to-protein traceability. To improve, we will expand the related works section (Page 2) to compare our methods explicitly: DNAMark leverages codon redundancy for DNA-level watermarking, while CentralMark extends to proteins via ESM embeddings, unlike Chen et al.’s protein-only focus. We will include more discussions in the revised paper.

---

> ### Author Response · Authors · 2025-08-07
>
> Dear reviewer,
>
> We hope our latest response has resolved your issue. If you are now satisfied, we would appreciate it if you would consider updating your support rating. Thank you for your feedback!
>
> Bests,
> Authors

---

### Note · Authors · 2025-08-13

We thank the AC and reviewers for their thoughtful feedback and constructive discussions. We would like to highlight both the novelty of our work and the extensive improvements made during the rebuttal period.

**Novelty and Contributions**:
Our work introduces DNAMark and CentralMark, the first watermarking methods for DNA generative models, addressing the urgent and emerging biosecurity risks of AI-generated synthetic DNA. This is **fundamentally different from watermarking in text or images**: DNA’s 4-token alphabet, biological constraints (e.g., codon usage bias, translation efficiency), and susceptibility to biological noise (mutations, synthesis/sequencing errors) pose unique challenges. CentralMark further enables **cross-dogma inheritance, allowing detection from DNA to protein—a capability absent in prior watermarking literature**. Our results reveal unique domain-specific findings, such as the failure of state-of-the-art text watermarking methods under DNA-specific attacks (e.g., synonymous codon substitutions).

**Rebuttal Improvements**:

**Functional/structural evaluation** on 10 diverse therapeutic proteins, showing preserved folds (TM-scores ~0.72 for DNAMark, ~0.68 for CentralMark) and <5% binding affinity loss via docking.

**Additional baselines**: Implemented Hu et al. (ICLR 2024) and Dathathri et al. (Nature 2024) and adapted them to DNA; also tested k-mer tokenization with GENERATOR.

**Ablation studies** on base-change strategies (1st/2nd/3rd position) justify our biologically informed design.

**Governance and adoption pathways** are proposed, covering regulatory mandates, synthesis provider incentives, and integration with IBBIS standards.

**Forgery resistance** detailed via key-seeded pseudorandom codon partitioning with future cryptographic extensions.

All technical details and extended results will be integrated into the final version to ensure clarity and reproducibility.

**Fit to NeurIPS**:

Our paper is an AI for Science contribution, blending ML innovation with domain-specific constraints in genomics and biosecurity. NeurIPS has a strong record of publishing interdisciplinary work (e.g., small molecule/protein generative models), and our results both advance the methodology of generative model watermarking and deliver actionable insights for biosecurity.

---

### Decision · Program_Chairs · 2025-09-17

**Decision:**

Accept (poster)

**Comment:**

This paper introduces two watermarking techniques, DNAMark and CentralMark, to address biosecurity challenges by enabling reliable tracking of engineered DNA sequences. DNAMark embeds watermarks through synonymous codon substitutions, maintaining the original biological function of the sequence. CentralMark extends this concept by designing inheritable watermarks that persist from DNA through protein translation, allowing for detection across the central dogma. Both approaches utilize semantic embeddings to improve robustness against natural mutations, synthesis errors, and adversarial attacks. Reviewers requested additional experiments to assess the method's performance with other baselines and its applicability to a broader range of tokenizers used in DNA generative models. Furthermore, Reviewers wanted to see a more comprehensive review of related work, which has to be included in the revised version of the paper. On the technical level, the proposed methods heavily rely on KGW [33] (the red/green watermarking). Many reviewers indicate that the paper has to be made more accessible to the larger NeurIPS community, e.g.: "Jargon and Logic Jumps in Embedding-Based Logit Perturbation" (Reviewer sX4o), or "The paper contains too many biological considerations that require verification from the biology community (e.g. whether TM-score correlates with function, codon translation efficiencies)" (Reviewer P59q). However, all Reviewers were positive about the submission, thus, I recommend acceptance.